# Graph Adaptive Autoregressive Moving Average Models

**Moshe Eliasof**[1]  **Alessio Gravina**[2]  **Andrea Ceni**[2]
**Claudio Gallicchio**[2]  **Davide Bacciu**[2]  **Carola-Bibiane Schönlieb**[1]

## Abstract

Graph State Space Models (SSMs) have recently been introduced to enhance Graph Neural Networks (GNNs) in modeling long-range interactions. Despite their success, existing methods either compromise on permutation equivariance or limit their focus to pairwise interactions rather than sequences. Building on the connection between Autoregressive Moving Average (ARMA) and SSM, in this paper, we introduce GRAMA, a Graph Adaptive method based on a learnable ARMA framework that addresses these limitations. By transforming from static to sequential graph data, GRAMA leverages the strengths of the ARMA framework, while preserving permutation equivariance. Moreover, GRAMA incorporates a selective attention mechanism for dynamic learning of ARMA coefficients, enabling efficient and flexible long-range information propagation. We also establish theoretical connections between GRAMA and Selective SSMs, providing insights into its ability to capture long-range dependencies. Experiments on 26 synthetic and real-world datasets demonstrate that GRAMA consistently outperforms backbone models and performs competitively with state-of-the-art methods.

## 1. Introduction

Graph learning (Scarselli et al., 2008; Micheli, 2009; Bruna et al., 2013; Defferrard et al., 2016; Kipf & Welling, 2016; Veličković et al., 2018) has become crucial in handling graph-structured data across various domains (Gravina & Bacciu, 2024), such as social networks (Kipf & Welling, 2016; Hamilton et al., 2017), molecular interactions (Xu et al., 2019; Bouritsas et al., 2022), and more (Khemani et al., 2024). The most popular framework of neural graph learning is that of Message Passing Neural Networks (MPNNs). Some prominent examples are GCN (Kipf & Welling, 2016), GAT (Veličković et al., 2018), GIN (Xu et al., 2019), and GraphConv (Morris et al., 2019). However, many MPNNs suffer from a critical shortcoming of *over-squashing* (Alon & Yahav, 2021; Di Giovanni et al., 2023), that hinders their ability to model long-range interactions. To address this limitation, several proposals were made, from graph rewiring (Topping et al., 2022; Di Giovanni et al., 2023; Karhadkar et al., 2023), to multi-hop MPNNs (Gutteridge et al., 2023), weight space regularization (Gravina et al., 2023; 2025), as well as Graph Transformers (GTs) (Yun et al., 2019; Dwivedi & Bresson, 2022; Kreuzer et al., 2021b). Specifically, GTs became popular because of their theoretical and often practical ability to capture long-range node interactions through the attention mechanism. However, the quadratic computational cost of full attention limits their scalability, and in some cases, they were found to underperform on long-range benchmarks when compared to standard MPNNs (Tönshoff et al., 2023).

At the same time, State Space Models (SSMs), such as S4 (Gu et al., 2021c) and Mamba (Gu et al., 2023), have emerged as promising, linear-complexity alternatives to Transformers. SSMs leverage a recurrent and convolutional structure to efficiently capture long-range dependencies while maintaining linear time complexity (Nguyen et al., 2023). Contemporary models like Mamba develop selective filters that prioritize context through input-dependent selection, offering compelling advantages in processing long sequences with reduced computational demands compared to transformers (Gu et al., 2023). Despite these benefits, adapting SSMs to the non-sequential structure of graphs remains a significant challenge. Perhaps the biggest challenge in applying SSMs to graph learning tasks, lies in the fundamental question of *"how to transform a graph into a sequence?"*. To this end, several graph SSM approaches were proposed, from a graph-to-sequence heuristic in Wang et al. (2024a), to studying the relationship between SSMs and spectral GNNs by pairwise interactions (Huang et al., 2024b), as well as sample-based random walk sequencing of

---

[1]Department of Applied Mathematics, University of Cambridge, Cambridge, United Kingdom [2]Department of Computer Science, University of Pisa, Pisa, Italy. Correspondence to: Moshe Eliasof <me532@cam.ac.uk>, Alessio Gravina <alessio.gravina@di.unipi.it>, Andrea Ceni <andrea.ceni@unipi.it>.

*Proceedings of the 42^{nd} International Conference on Machine Learning*, Vancouver, Canada. PMLR 267, 2025. Copyright 2025 by the author(s).

the graph (Behrouz & Hashemi, 2024). More broadly, this question has also been studied in other, non-SSM related works, discussed in Appendix A. However, as we discuss later, some of them lose the permutation-equivariance property desired in GNNs, while others do not take advantage of the sequence processing ability of SSMs. These limitations hinder their ability to fully leverage sequence-processing capabilities, especially for addressing oversquashing in GNNs. To resolve these issues, we propose, instead, a complementary approach – transforming a static input graph into a sequence of graphs, combined with an adaptive neural autoregressive moving-average (ARMA) mechanism, called GRAMA. We show that GRAMA is theoretically equivalent to an SSM on graphs. Our GRAMA allows us to enjoy the benefits of sequential processing mechanisms like SSMs, coupled with any GNN backbone, from MPNNs to graph transformers, while maintaining backbone properties, such as permutation-equivariance.

**Main Contributions.** Our Adaptive **Gr**aph **A**utoregressive **M**oving **A**verage (GRAMA) model offers several advancements in the conjoining of dynamical systems theory into GNNs:

- **Principled Integration of SSMs in GNNs.** We enable the use of sequence-based models (like ARMA) coupled with virtually any GNN backbone, by transforming graph inputs into temporal sequences without sacrificing permutation invariance.

- **Theoretical Understanding of the coupling of SSMs and GNNs.** We demonstrate that augmenting GNNs with ARMA via our GRAMA has an equivalent SSM model.

- **Mitigation of the oversquashing problem.** We provide the theoretical foundation that our GRAMA effectively addresses the oversquashing phenomenon in GNNs and improves the long-range interaction modeling capabilities.

- **Strong Practical Performance.** We demonstrate our GRAMA on three popular backbones (GCN (Kipf & Welling, 2016), GatedGCN (Bresson & Laurent, 2018), and GPS (Rampášek et al., 2022)) and show the compelling performance by GRAMA on 26 synthetic and real-world datasets.

## 2. Related Work

We now provide an overview and discussion of related topics and works to our GRAMA. In Appendix A, we discuss additional related works.

**Long-Range Interactions on Graphs.** GNNs rely on message-passing mechanisms to aggregate information from neighboring nodes, which limits their ability to capture long-range dependencies, as highlighted by Alon & Yahav (2021); Di Giovanni et al. (2023). Models like GCN (Kipf

& Welling, 2016), GraphSAGE (Hamilton et al., 2017), and GIN (Xu et al., 2019) face challenges such as over-smoothing (Nt & Maehara, 2019; Oono & Suzuki, 2020; Cai & Wang, 2020; Rusch et al., 2023), over-squashing (Alon & Yahav, 2021; Topping et al., 2022; Di Giovanni et al., 2023) and more generally vanishing gradients (Arroyo et al., 2025), which hinder long-range information propagation—critical in applications like bioinformatics (Baek et al., 2021; Dwivedi et al., 2022b) and heterophilic settings (Luan et al., 2024; Wang et al., 2024b). To address these limitations, various methods have emerged, including graph rewiring (Topping et al., 2022; Karhadkar et al., 2023), adaptive message passing (Errica et al., 2024; Finkelshtein et al., 2024), weight space regularization (Gravina et al., 2023; 2024b; 2025), exploitation of port-Hamiltonian dynamics (Heilig et al., 2025), and Graph Transformers (GTs). GTs, which capture both local and global interactions, have been particularly promising, as demonstrated by models like SAN (Kreuzer et al., 2021c), Graphormer (Ying & Leskovec, 2021), and GPS (Rampášek et al., 2022). These models often incorporate positional encodings, such as Laplacian eigenvectors (Dwivedi et al., 2021) or random-walk structural encodings (RWSE) (Dwivedi et al., 2022a), to encode graph structure. However, the quadratic complexity of full attention in GTs presents scalability challenges. Recent innovations like sparse attention mechanisms (Zaheer et al., 2020; Choromanski et al., 2020), Exphormer (Shirzad et al., 2023), and linear graph transformers (Wu et al., 2023; Deng et al., 2024) address these bottlenecks, improving efficiency and scalability for long-range propagation.

**State Space Models (SSMs).** SSMs, traditionally used for time series analysis (Hamilton, 1994b; Aoki, 2013), process sequences through latent states. However, classic SSMs struggle with long-range dependencies and lack parallelism, limiting their computational efficiency. Recent advances, such as the Structured State Space Sequence model (S4) (Gu et al., 2021c; Fu et al., 2023), mitigate these issues by employing linear recurrence as a structured convolutional kernel, enabling parallelization on GPUs. Despite this, simple SSMs still underperform compared to attention models in natural language tasks. Mamba (Gu et al., 2023) improves the ability of SSMs to capture long-range dependencies by selectively controlling which sequence parts influence model states. Mamba has shown promising results, outperforming Transformers in several benchmarks (Gu et al., 2023; Liu et al., 2024) while being more computationally efficient. The combination of SSMs with graph models presents challenges, particularly in transforming the articulated connectivity of graphs into sequences. For instance, Graph-Mamba (Wang et al., 2024a) orders nodes by degree, but this heuristic approach sacrifices permutation-equivariance, a desirable property in GNNs. Similarly, Behrouz & Hashemi (2024) propose generating

sequences via random walks, which improves performance but also sacrifices permutation-equivariance while adding non-determinism to the model. Also, turning a graph into a sequence based on a policy, such as sorting nodes by degree, limits direct use of the input graph, as multiple graphs can share the same node degrees and thus be indistinguishable. Huang et al. (2024b) explored links between spectral GNNs and graph SSMs, focusing on pairwise interactions; however, this design choice may not fully exploit the sequence-handling capacity of SSMs and may reach the state of oversquashing earlier because of the use of powers of the adjacency matrix (Di Giovanni et al., 2023). In this work, we harness the potential of SSMs by adopting a structure inspired by the connection between SSMs and ARMA models. By transforming static graphs into sequences, GRAMA maintains permutation-equivariance, a desired property in GNNs (Bronstein et al., 2021), also useful for long-propagation (Pan & Kondor, 2022; Schatzki et al., 2024), while enabling effective learnable and selective long-range propagation.

**Autoregressive Moving Average Models (ARMA).** ARMA models, introduced by Whittle (1951), combine an autoregressive (AR) component, modeling dependencies on previous time steps, with a moving average (MA) component, considering residuals. Widely applied in stationary time series analysis (Box et al., 1970), ARMA models are equivalent to state space models (SSMs) (Hamilton, 1994a). An ARMA$(p, q)$ model considers previous $p$ states and $q$ residuals $\delta(\cdot)$, and is governed by the following equation:

$$f(t) = \sum_{i=1}^{p} \phi_i f(t-i) + \sum_{j=1}^{q} \theta_j \delta(t-j) + \delta(t), \quad (1)$$

where $\{\phi_i\}_{i=1}^{p}$, $\{\theta_i\}_{j=1}^{q}$ are the autoregressive and moving average coefficients, respectively.

Although ARMA models are traditionally used for processing sequences, they have also been studied for classical graph filtering (Isufi et al., 2016) and more recently formulated as an MPNN in Bianchi et al. (2021). The Graph ARMA model (Bianchi et al., 2021) introduced a learnable ARMA version for GCNs, using recursive 1-hop filters to create a structure resembling ARMA methods. In this paper, we introduce GRAMA, a method that leverages neural ARMA models by transforming a static input graph into a graph sequence. Different than Bianchi et al. (2021), which uses the static input graph and formulates a recursive ARMA model through a spectral convolution perspective, our GRAMA incorporates a selective and graph adaptive mechanism that learns ARMA coefficients along the graph sequence. This dynamic adjustment of coefficients directly addresses oversquashing by preserving long-range dependencies and enabling adaptive control over feature propagation. Additionally, Bianchi et al. (2021) uses an ARMA$(1, 1)$ model with non-linearities between steps, hin-

dering its direct conversion into an SSM, while we show that our GRAMA has an equivalent SSM, providing deeper theoretical understandings.

## 3. GRAMA

Although a graph is a static structure, the process of message passing introduces a dynamic element. In message passing, information is propagated through the graph, allowing nodes to update their states based on the states of their neighbors. This dynamic behavior can be viewed through the lens of dynamical systems, where the state of each node evolves according to certain aggregating rules, as discussed in Section 2. This perspective is instrumental in Recurrent Neural Networks (RNNs), which are designed to handle sequential data and capture temporal dependencies. By treating the message-passing process as a dynamical system, we can leverage the strengths of RNNs to model the evolution of node states over time. The model we propose, GRAMA, takes inspiration from the architectural structure of the latest generation of sequential models, like S4 (Gu et al., 2021a), Mamba (Gu et al., 2023), LRU (Orvieto et al., 2023b), and xLSTM (Beck et al., 2024). To import these powerful sequential models to graph learning, we first translate static input graphs into sequences of graphs. Then, the GRAMA block transforms such graph sequence into another graph sequence, while considering the structure of the graph. Each GRAMA block is linear, and non-linear activations are applied between GRAMA blocks to increase the flexibility of the overall model. Below, we discuss in detail the different aspects of our GRAMA – from its initialization to the graph sequence processing blueprint by ARMA, to the learning of ARMA coefficients in a graph adaptive manner. The overall design of GRAMA is illustrated in Figure 1.

**Notations.** We denote a graph by $G = (V, E)$, with $|V| = n$ nodes and $|E| = m$ edges. A node $v$ is associated with input node features $f_v \in \mathbb{R}^c$. The node features are then denoted by $\mathbf{f} \in \mathbb{R}^{n \times c}$.

**Initialization.** Processing information with ARMA or SSM frameworks, by design, requires a sequence. As discussed in Section 2, previous studies on graph SSMs have chosen to transform the graph into a sequence by means of heuristic node ordering, random walk sampling, or by considering pairwise interactions (edges) as sequences of length 2. While these choices are valid, and show strong performance in practice, they also introduce challenges compared to common graph learning approaches, or may not fully utilize the underlying sequence processing framework. Specifically, the first two approaches (node ordering and walk sampling) do not maintain the permutation equivariance desired in GNNs, and the third (pairwise interactions) considers only very short sequences, while one key benefit of the ARMA and SSM frameworks is their ability to cap-

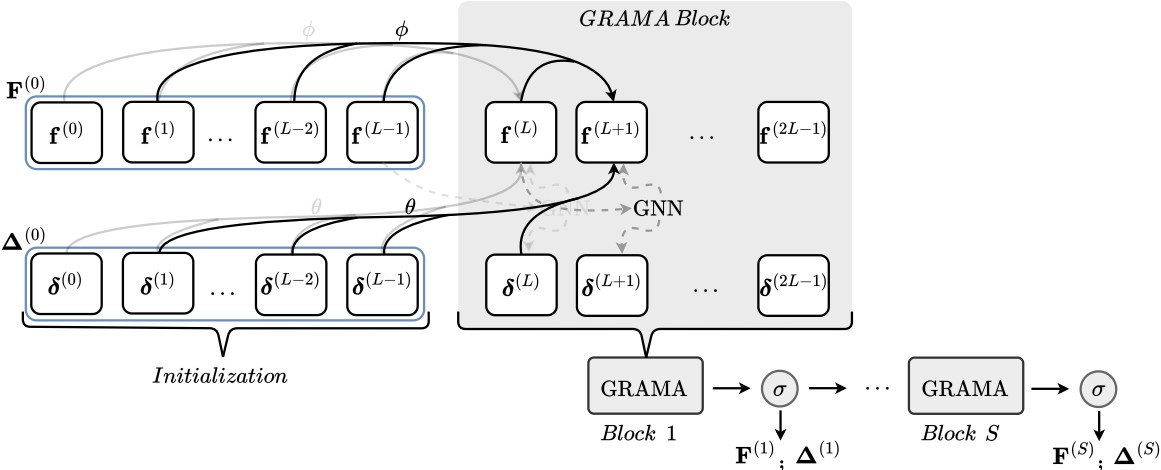

Figure 1: An illustration of the GRAMA framework with $L$ recurrences. We embed a static input graph into a sequence of graphs. This sequence is the input for the first GRAMA block. Here, a GRAMA block is composed of a neural ARMA$(L, L)$ layer with adaptive autoregressive $\phi = \{\phi_i\}_{i=1}^{L}$ and moving average $\theta = \{\theta_j\}_{j=1}^{L}$ coefficients, that weigh previous states $\{\mathbf{f}_l\}_{l=0}^{L-1}$ and residuals $\{\delta_l\}_{l=0}^{L-1}$, and a graph-informed residual update via a GNN backbone. A GRAMA block yields two updated state and residual sequences $\mathbf{F}^{(s)}, \boldsymbol{\Delta}^{(s)}$ for the $s = 1, \ldots, S$ block. Each GRAMA block is a linear system, and non-linearities are applied between GRAMA blocks, as in Equation (8).

ture long-range dependencies in long sequences (Gu et al., 2021b). To address these challenges, we propose to transform a *static graph* into a *sequence of graphs*, such that each node is equipped with a sequence of input node feature vectors rather than a single input node feature vector. By following this idea, we can employ sequence processing frameworks such as ARMA on data beyond pairwise interactions, while maintaining permutation-equivariance, as we discuss later. Specifically, we first stack the input node features $\mathbf{f}$ for $L$ times, where $L > 0$ is a hyperparameter that determines the length of the sequence to process, followed by the application of a set of MLPs, $\{g_k\}_{k=0}^{L-1}$, one for each $k = 0, \ldots, L - 1$, that embed the original $c$ node features into $d$ channels:

$$\mathbf{F}^{(0)} = \left[\mathbf{f}^{(0)}, \ldots, \mathbf{f}^{(L-1)}\right] = \left[g_0(\mathbf{f}), \ldots, g_{L-1}(\mathbf{f})\right], \quad (2)$$

where $\mathbf{F}^{(0)} \in \mathbb{R}^{L \times n \times d}$. We refer to the sequence encoded by $\mathbf{F}^{(0)}$ as the initial input sequence, and to work with an ARMA model, we also define the *residuals* sequence as follows:

$$\boldsymbol{\Delta}^{(0)} = \left[\boldsymbol{\delta}^{(0)}, \ldots, \boldsymbol{\delta}^{(L-1)}\right], \quad \boldsymbol{\Delta}^{(0)} \in \mathbb{R}^{L \times n \times d}, \quad (3)$$

where $\boldsymbol{\delta}^{(\ell)} = \mathbf{f}^{(\ell+1)} - \mathbf{f}^{(\ell)}$ for $\ell = 0, \ldots, L - 2$. Note that by subtracting subsequent elements in the input sequence $\mathbf{F}^{(0)}$, we are left with $L - 1$ elements. Therefore, we choose the last residual term in $\boldsymbol{\Delta}^{(0)}$ (that is $\boldsymbol{\delta}^{(L-1)}$) to be a matrix of zeros at the initialization step.

We note that via this approach, we can perform sequence modeling using ARMA on the sequence dimension ($L$) while retaining the ability to use any desired backbone GNN

to exchange information between nodes, as shown in Section 3.1, thus rendering our GRAMA a drop-in mechanism.

### 3.1. Graph Neural ARMA

**Autoregressive (AR) Layers.** An AR$_p$ captures the relationship between current node features and their $p > 0$ previous historical values, through the learnable coefficients $\{\phi_i\}_{i=1}^{p}$ discussed in Section 3.2. Formally, given a sequence of node features of length $L$ $\left[\mathbf{f}^{(\ell)}, \ldots, \mathbf{f}^{(\ell+L-1)}\right]$, assuming $p \le L$, the node features at step $\ell + L$ read:

$$\mathbf{f}_{\text{AR}_p}^{(\ell+L)} = \text{AR}_p(\mathbf{f}^{(\ell)}, \ldots, \mathbf{f}^{(\ell+L-1)}) = \sum_{i=1}^{p} \phi_i \mathbf{f}^{(\ell+L-i)}. \quad (4)$$

**Moving Average (MA) Layers.** Given a residuals sequence $\left[\boldsymbol{\delta}^{(\ell)}, \ldots, \boldsymbol{\delta}^{(\ell+L-1)}\right]$, a MA$_q$ layer with $\{\theta_j\}_{j=1}^{q}$ learnable coefficients, captures the dependency of the latest $0 < q \le L$ residuals:

$$\mathbf{f}_{\text{MA}_q}^{(\ell+L)} = \text{MA}_q(\boldsymbol{\delta}^{(\ell)}, \ldots, \boldsymbol{\delta}^{(\ell+L-1)}) = \sum_{j=1}^{q} \theta_j \boldsymbol{\delta}^{(\ell+L-j)}. \quad (5)$$

**GRAMA Recurrence.** Combining AR$_p$ and MA$_q$ layers, leads to the ARMA$(p, q)$ recurrence:

$$\mathbf{f}^{(\ell+L)} = \mathbf{f}_{\text{AR}_p}^{(\ell+L)} + \mathbf{f}_{\text{MA}_q}^{(\ell+L)} + \boldsymbol{\delta}^{(\ell+L)}, \quad (6)$$

where $\boldsymbol{\delta}^{(\ell+L)}$ is the residual of the last step, which is given by a GNN backbone that is optimized jointly with the ARMA coefficients, that is, $\boldsymbol{\delta}^{(\ell+L)} = \text{GNN}(\mathbf{f}^{(\ell+L-1)}; G)$.

Here, we apply the GNN backbone without non-linearity so that each recurrence step within a GRAMA block is a linear function. In particular, note that the GNN can be any graph neural network, because at each recurrence, GRAMA processes a sequence of graphs by updating each node feature based on its sequence via the terms $\mathbf{f}_{\mathrm{AR}_p}^{(\ell+L)}, \mathbf{f}_{\mathrm{MA}_q}^{(\ell+L)}$, coupled a with a GNN in the term $\boldsymbol{\delta}^{(\ell+L)}$. Moreover, the structure of the terms $\mathbf{f}_{\mathrm{AR}_p}^{(\ell+L)}, \mathbf{f}_{\mathrm{MA}_q}^{(\ell+L)}$ includes multiple residual connections, which can implement standard skip-connections, retaining the expressiveness of the backbone GNN. Section 5 showcases GRAMA with various GNN backbones, from MPNNs to graph transformers.

**GRAMA Block.** Equation (6) describes a *single* recurrence step within a GRAMA block. Similar to other recurrent mechanisms, we apply $R$ recurrences, where $R > 1$ is a hyperparameter. Thus, given the initial states $\mathbf{F}^{(0)}$ and residuals $\boldsymbol{\Delta}^{(0)}$, after $R$ recurrences according to Equation (6), we obtain updated states $\left[\mathbf{f}^{(L)}, \ldots, \mathbf{f}^{(L+R-1)}\right]$ and residuals $\left[\boldsymbol{\delta}^{(L)}, \ldots, \boldsymbol{\delta}^{(L+R-1)}\right]$ sequences, followed by an element-wise application of non-linearity $\sigma$:

$$\begin{aligned} \mathbf{F}^{(1)} &= \left[\sigma(\mathbf{f}^{(L)}), \ldots, \sigma(\mathbf{f}^{(L+R-1)})\right], \\ \boldsymbol{\Delta}^{(1)} &= \left[\sigma(\boldsymbol{\delta}^{(L)}), \ldots, \sigma(\boldsymbol{\delta}^{(L+R-1)})\right]. \end{aligned} \quad (7)$$

In practice, as discussed in Appendix D.6, $R$ is chosen such that $p = q = R = L$, and the obtained updated sequences are $\mathbf{F}^{(1)} = \left[\sigma(\mathbf{f}^{(L)}), \ldots, \sigma(\mathbf{f}^{(2L-1)})\right]$, $\boldsymbol{\Delta}^{(1)} = \left[\sigma(\boldsymbol{\delta}^{(L)}), \ldots, \sigma(\boldsymbol{\delta}^{(2L-1)})\right]$.

**Deep GRAMA.** In Equation (7), we describe the action of a *single, first* GRAMA block. Overall, each block performs $R$ recurrence steps. As such, the first GRAMA block yields $R$ new states and residuals encoded in $\mathbf{F}^{(1)}$ and $\boldsymbol{\Delta}^{(1)}$, respectively, that can then be processed by subsequent GRAMA blocks. That is, we can stack $S \geq 1$ GRAMA blocks, each block with its own parameters, forming a deep GRAMA network, where the updated sequences at the $s$-th GRAMA block are:

$$\begin{aligned} \mathbf{F}^{(s)} &= \left[\sigma(\mathbf{f}^{(L+(s-1)R)}), \ldots, \sigma(\mathbf{f}^{(L+sR-1)})\right], \\ \boldsymbol{\Delta}^{(s)} &= \left[\sigma(\boldsymbol{\delta}^{(L+(s-1)R)}), \ldots, \sigma(\boldsymbol{\delta}^{(L+sR-1)})\right], \end{aligned} \quad (8)$$

for $s = 1, \ldots, S$. Note that the depth of a GRAMA network is therefore equivalent to the number of systems $S$ to be learned, multiplied by the number of recurrent steps $R$. The outputs of the GRAMA network are then the final state and residual sequences $\mathbf{F}^{(S)}, \boldsymbol{\Delta}^{(S)}$. We illustrate this process in Figure 1. Because in our experiments we are interested in static graph learning problems, we feed the latest state matrix within the sequence $\mathbf{F}^{(S)}$ to a readout layer to obtain the final prediction, as elaborated in Appendix C.2. The additional processing in GRAMA introduces some computational overhead, as detailed in Section 3.3. However, this cost remains reasonable compared to other methods and yields significant performance improvements, as detailed in Section 5.

### 3.2. Learning Adaptive Graph ARMA Coefficients

We now introduce our graph adaptive approach for learning the ARMA coefficients, which is a key component in our approach to allow a flexible and selective GRAMA.

**Naive ARMA Learning.** The most straightforward way to learn the AR and MA coefficients, $\{\phi_i\}_{i=1}^p$ and $\{\theta_j\}_{j=1}^q$, is to consider them as parameters of the neural network and learn them via gradient descent. However, this yields coefficients that are identical for all inputs, thereby not adaptive. This approach is directly linked to non-selective weights in SSM models (Gu et al., 2021c), which were shown to be less effective compared to selective coefficients (Gu et al., 2023).

**Selective ARMA Learning.** To allow selective ARMA coefficient learning similarly to Mamba (Gu et al., 2023), we use an attention mechanism (Vaswani et al., 2017) applied over the state and residual sequences $\mathbf{F}^{(s)}, \boldsymbol{\Delta}^{(s)}$ at each GRAMA block $s = 1, \ldots, S$. The rationale behind this construction is that an attention layer assigns scores between elements within the sequence. Formally, we obtain two scores matrices $\mathcal{A}_{\mathbf{F}^{(s)}}, \mathcal{A}_{\boldsymbol{\Delta}^{(s)}} \in [0, 1]^{L \times L}$. The last row in each matrix represents the predicted coefficients for our GRAMA, $\{\phi_i\}_{i=1}^p$ and $\{\theta_j\}_{j=1}^q$, respectively. However, the SoftMax normalization in standard attention layers yields non-negative pairwise values, which is not consistent with the usual choice of ARMA coefficients. Therefore, we follow the self-attention implementation (Vaswani et al., 2017) up to the SoftMax step, and we normalize the scores to be in $[-1, 1]$ while complying with a sum-to-one constraint. We note that, this procedure facilitates learning stability, such that ARMA coefficients do not explode or vanish, and its design is guided by the insights from Theorems 4.3 and 4.4. We also note that this overall construction yields two-fold adaptivity in the predicted ARMA coefficients: First, the attention mechanism allows selectivity with respect to inputs, which are the sequences $\mathbf{F}^{(s)}, \boldsymbol{\Delta}^{(s)}$. Second, because these sequences are coupled with a GNN backbone, as shown in Equation (6), it implies that the input node features and the graph structure influence the coefficients. We provide further implementation details in Appendix C, and a comparison between naive and selective ARMA learning in Appendix E.4.

### 3.3. Time and Space Complexity of GRAMA

We discuss the theoretical complexity of our method, showing its reduced computational complexity compared with other models, e.g., transformers. We note that, overall, our GRAMA retains the asymptotic complexity of the underlying GNN backbone, assuming that the number of recurrences is a constant, or smaller than the number of nodes and edges within the graph. We report empirical runtimes in Appendix E.3, demonstrating that GRAMA offers better

scalability and performance compared to other approaches such as graph transformers.

**Time Complexity.** We analyze the case where we use an MPNN that is linear in graph size (nodes and edges) such as GCN is used within GRAMA. Our GRAMA is comprised of $L$ initial MLPs, $S$ GRAMA blocks, each with $L$ recurrent steps, and a final readout layer. Note that $L$ is the sequence length, which is a hyperparameter and does not exceed the value of 50 in our experiments. The initial MLPs operate on the input $f \in \mathbb{R}^{n \times c}$ and embed them to a hidden dimension $d$. Therefore, their time complexity is $\mathcal{O}(L \cdot n \cdot c \cdot d)$. Each GRAMA block is comprised of two attention layers – one for the *pooled* states sequence and the other for the residual *pooled* sequence, and a GNN layer for predicting the current step residual, which operates on graph node features. The attention layer time complexity is $\mathcal{O}(L^2 d^2)$ because the pooled (across the graph nodes) sequence is of shape $L \times d$, and the GNN layer complexity is $\mathcal{O}(n + m)$, where $n$ is the number of nodes and $m$ is the number of edges in the graph. Note that usually $m \gg n$, so the GNN complexity can be rewritten as $\mathcal{O}(m)$. We note that, in the case of a graph-transformer based GNN, like GPS, we have that $m = n^2$. In the following, we consider the more general case. In total, we have $S$ GRAMA blocks, where $S$ is a hyperparameter, and is typically small, up to 4. The final readout layer is a standard MLP and, therefore, has the time complexity of $\mathcal{O}(n \cdot d \cdot o)$ for node-wise tasks, and $\mathcal{O}(d \cdot o)$ for graph-level tasks. Therefore, the overall time complexity (including initial MLPs and readout) of our GRAMA is $\mathcal{O}\left(L \cdot n \cdot c \cdot d + SL \cdot (n + m + L^2 \cdot d^2) + n \cdot d \cdot o\right)$.

**Space Complexity.** We analyze the case where linear in graph size (nodes and edges) complexity MPNN (such as GCN) is used within GRAMA. The space complexity of the initial MLPs is $\mathcal{O}(L \cdot c \cdot d)$. The space complexity for each GRAMA block is $\mathcal{O}(d^2)$ for the GNN layer, and similarly $\mathcal{O}(d^2)$ for the two attention layers. Overall, we have $S$ such blocks. The readout layer space complexity is $\mathcal{O}(d \cdot o)$. Thus, the overall space complexity (including initial MLPs and readout) of GRAMA is $\mathcal{O}(L \cdot c \cdot d + S \cdot d^2 + d \cdot o)$.

## 4. Theoretical Properties of GRAMA

We now formally cast common knowledge formulated in the context of RNNs, control theory, and SSMs (Yu et al., 2019; Slotine et al., 1991; Khalil, 2002; Aoki, 2013) to the realm of GNNs, aiming to adapt foundational results from non-graph settings of SSMs and ARMA models into a graph-learning framework. We discuss the main theoretical properties of our GRAMA: (i) its representation as an SSM model, (ii) its stability, and (iii) its ability to model long-range interactions in graphs. All the proofs are provided in Appendix B.

**Connection to SSM.** As discussed in Section 3, each GRAMA block is fundamentally an ARMA model. In Theorem 4.1, we formalize the equivalence between ARMA models and linear SSMs. This allows us to interpret our GRAMA model as a stack of graph-informed SSMs through the backbone GNN encoded in Equation (6).

**Theorem 4.1** (Equivalence between ARMA models and State Space Models). *For every ARMA model, there exists an equivalent State Space Model (SSM) representation, and conversely, for every linear SSM, there exists an equivalent ARMA model representation.*

**Stability.** Representing an ARMA system as an SSM involves the description of a linear recurrence equation as $\mathbf{f}^{(L)} = \sum_{i=1}^{p} \phi_i \mathbf{f}^{(L-i)} + \sum_{j=1}^{q} \theta_j \boldsymbol{\delta}^{(L-j)} + \boldsymbol{\delta}^{(L)}$, or, alternatively, in matrix form as $\mathbf{X}^{(L)} = \mathbf{A}\mathbf{X}^{(L-1)} + \mathbf{B}\boldsymbol{\delta}^{(L)}$, with $\mathbf{X}^{(L-1)} = \left[\mathbf{f}^{(L-1)}, \dots, \mathbf{f}^{(0)}, \boldsymbol{\delta}^{(L-1)}, \dots, \boldsymbol{\delta}^{(0)}\right]$, see Appendix B for more details.[1] In the SSM literature, the $\mathbf{A}$ matrix is called the *state matrix*. The state matrix corresponding to a GRAMA block is entirely determined by the set of autoregressive and moving average coefficients. Thus, each GRAMA block is characterized by an adaptive state matrix, which is especially important since it directly governs the evolution of the node features $\mathbf{f}$. In particular, the stability of this evolution can be established by analyzing the powers of the state matrix, as widely studied in the context of RNN and SSM theory (Pascanu, 2013; Gu et al., 2021b). Hence, the stability of a GRAMA block can be characterized by the following Lemma 4.2.

**Lemma 4.2** (Stability of GRAMA). *The linear SSM corresponding to a GRAMA block with autoregressive coefficients $\{\phi_i\}_{i=1}^{p}$ is stable if and only if the spectral radius of its state matrix is less than (or at most equal to) 1. In particular, this happens if and only if the polynomial $P(\lambda) = \lambda^p - \sum_{j=1}^{p} \phi_j \lambda^{p-j}$ has all its roots inside (or at most on) the unit circle.*

We now give a sufficient condition for the stability of the SSM corresponding to a GRAMA block.

**Theorem 4.3** (Sufficient condition for GRAMA stability). *If $\sum_{j=1}^{p} |\phi_j| \leq 1$, then the GRAMA block with autoregressive coefficients $\{\phi_i\}_{i=1}^{p}$ corresponds to a stable linear SSM.*

**Long-Range Interactions.** A key distinction between standard MPNNs and our GRAMA lies in its neural selective sequential mechanism, which uses learned ARMA coefficients to operate across two domains: the spatial graph domain via a GNN backbone, and the sequence domain via the ARMA mechanism, enabling selective state updates. Remarkably,

---

[1]Note that, following the notation of Section 3, we can write the state $\mathbf{X}^{(L-1)}$ as the concatenation of $\mathbf{F}^{(0)}$ and $\boldsymbol{\Delta}^{(0)}$, i.e., $\mathbf{X}^{(L-1)} = \left[\mathbf{F}^{(0)}, \boldsymbol{\Delta}^{(0)}\right]$. For simplicity of notations, we analyze the case where $R = L$.

the state matrices of each GRAMA block play a significant role in the propagation of the information from the first sequence of node features, $\mathbf{F}^{(0)} = \left[\mathbf{f}^{(0)}, \ldots, \mathbf{f}^{(L-1)}\right]$, to the last sequence of node features after $S$ GRAMA blocks, $\mathbf{F}^{(S)} = \left[\mathbf{f}^{(LS)}, \ldots, \mathbf{f}^{(L(S+1)-1)}\right]$, especially for large $L$ and $S$. In fact, if the entries of the $k$-th power of the state matrix of a GRAMA block vanish, then for a stable GRAMA it is impossible to model long-range dependencies of $k$ hops, in the sequence, as we show in Lemma B.1.

This fact relates to a broadly acknowledged problem in the RNN literature, the vanishing gradient issue (Hochreiter et al., 2001; Bengio et al., 1994; Orvieto et al., 2023a): the entries of the powers of a matrix with a spectral radius less than 1 can quickly vanish, making it challenging for gradient-based algorithms to effectively long-range patterns. Therefore, to bias the long-term propagation of the information of a GRAMA block, we can initialize the state matrix to have its eigenvalues close enough to the unitary circle, following the footsteps of recent RNN methodologies (Orvieto et al., 2023b; Arjovsky et al., 2016; De et al., 2024). In fact, the closer the eigenvalues are to the unitary circle, the slower the powers of the state matrix vanish (Horn & Johnson, 2012). The following Theorem 4.4 provides a criterion to control the long-range interaction of GRAMA.

**Theorem 4.4** (GRAMA allows long-range interactions). *Let us be given a* GRAMA *block with autoregressive coefficients* $\{\phi_i\}_{i=1}^{p}$. *Assume the roots of the polynomial* $P(\lambda) = \lambda^p - \sum_{j=1}^{p} \phi_j \lambda^{p-j}$ *are all inside the unit circle. Then, the closer the roots* $P(\lambda)$ *are to the unit circle, the longer the range propagation of the linear SSM corresponding to such a* GRAMA *block.*

The results derived in this section provide the theoretical foundation and motivation for the employment of GRAMA as a method to address the oversquashing phenomenon in GNNs, and to enhance long-range interaction modeling capabilities, as we show in our experiments in Section 5.

# 5. Experiments

We present the empirical performance of our GRAMA on a suite of benchmarks similar to previous graph SSM studies. Specifically, we show the efficacy in performing long-range propagation, thereby mitigating oversquashing. To this end, we evaluate GRAMA on a graph transfer task (Gravina et al., 2025) in Section 5.1. In a similar spirit, we assess GRAMA on synthetic benchmarks that require the exchange of messages at large distances over the graph, called graph property prediction from Gravina et al. (2023), in Section 5.2. We also verify GRAMA on real-world datasets, including the long-range graph benchmark (Dwivedi et al., 2022b) in Section 5.3, and additional GNN benchmarks in Appendix E.1, where we consider MalNet-Tiny (Freitas et al., 2021), the heterophilic node

classification datasets from Platonov et al. (2023), ZINC-12k, OGBG-MOLHIV, Cora, CiteSeer, Pubmed, MNIST CIFAR10, PATTERN, and CLUSTER. In Appendix E.3, we discuss the runtimes of GRAMA, and compare with other methods. In Appendix E.4, we report ablation studies and additional comparisons to provide a comprehensive understanding of our GRAMA, while in Appendix E.6 we include an evaluation on temporal setting. Notably, the performance of GRAMA is compared with popular and state-of-the-art methods, such as MPNN-based models, DE-GNNs, higher-order GNNs, and graph transformers, and shows consistent improvements over its baseline models, with competitive results to state-of-the-art methods (see Appendix F). We note that, in the main text, we report models and variants that are state-of-the-art on the individual benchmarks, which may lead to differences between the tables, while more variants are explored in the appendix. Additional details on baseline methods are presented in Appendix D.1, and the explored grid of hyperparameters in Appendix D.6. We demonstrate GRAMA on three widely used backbones—GCN (Kipf & Welling, 2016), GatedGCN (Bresson & Laurent, 2018), and GPS (Rampášek et al., 2022), highlighting its versatility across different backbone types, including linear MPNNs and graph transformers, and its consistently strong performance regardless of the underlying backbone architecture. We release our code at `https://github.com/MosheEliasof/GRAMA`.

## 5.1. Graph Feature Transfer

**Setup.** We consider three graph feature transfer tasks based on (Gravina et al., 2025). The objective is to transfer a label from a source to a target node, placed at a distance $\ell$ in the graph. By increasing $\ell$, we increase the complexity of the task and require longer-range information. Moreover, due to oversquashing, the performance is expected to degrade as $\ell$ increases. We initialize nodes with a random valued feature, and we assign values "1" and "0" to source and target nodes, respectively. We consider three graph distributions, i.e., line, ring, crossed-ring, and four different distances $\ell = \{3, 5, 10, 50\}$. Appendix D.2 provides additional details about the dataset and the task.

**Results.** Figure 2 reports the test mean-squared error (and standard deviation) of GRAMA compared to well-known models from the literature. Results show that traditional MPNNs (GCN, GAT, GraphSAGE, and GIN) struggle to propagate information effectively over long distances, with their performance deteriorating significantly as the source-target distance $\ell$ increases. This is evident across all graph types. In contrast, GRAMA coupled with GCN achieves a low error even when the source-target distance is 50. Among the models, A-DGN, SWAN, and GPS come closest to GRAMA performance, as they are a non-dissipative approach and a transformer-based model, respectively. How-

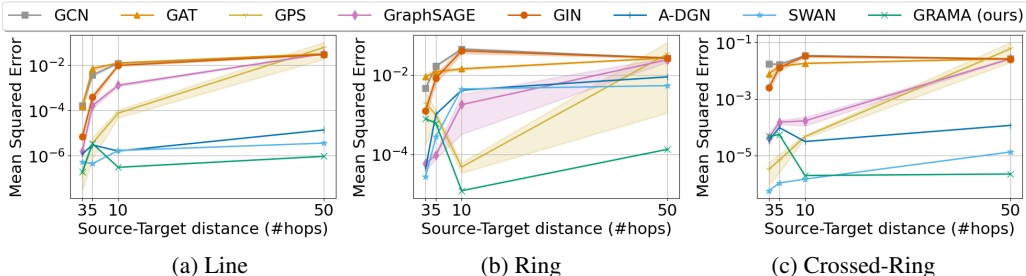

Figure 2: Feature transfer performance on (a) Line, (b) Ring, and (c) Crossed-Ring graphs.

ever, GRAMA still outperforms all baselines across all graph structures, especially as the propagation distance increases, thereby offering solid empirical evidence of its ability to transfer information across long distances, as supported by our theoretical understanding from Section 4.

### 5.2. Graph Property Prediction

**Setup.** We consider the three graph property prediction tasks presented in (Gravina et al., 2023), investigating the performance of GRAMA in predicting graph diameters, single source shortest paths (SSSP), and node eccentricity on synthetic graphs. To effectively address these tasks, it is essential to propagate information not only from direct neighbors but also from distant nodes within the graph. As a result, strong performance in these tasks mirrors the ability to facilitate long-range interactions. We provide more details on the setup and task in Appendix D.3. For the GPS results, we use a basic GPS with no additional components (e.g., encodings), to quantify the contribution of GRAMA.

**Results.** Table 1 reports the mean test $log_{10}(\text{MSE})$, comparing our GRAMA with various MPNNs, DE-GNNs, and transformer-based models. The results highlight that GRAMA$_{\text{GPS}}$ consistently achieves the best performance across all tasks, demonstrating significant improvements over baseline models. For example, in the Eccentricity task, GRAMA$_{\text{GPS}}$ reduces the error score by over 1.2 points compared to SWAN and by over 1.7 points compared to A-DGN, which are models designed to propagate information over long radii effectively. Compared to ARMA (Bianchi et al., 2021), our method demonstrates an average improvement of 2.4 points, highlighting the empirical difference between the methods, besides their major qualitative differences.

Overall, these results further validate the effectiveness of our GRAMA in modeling long-range interactions and mitigating oversquashing. Furthermore, GRAMA not only surpasses strong models like GPS, but also strengthens the performance of simple MPNN backbones like GCN. For example, GCN augmented with our GRAMA consistently delivers better results than the baseline GCN, highlighting its ability to enhance traditional message-passing frameworks.

Table 1: Mean test set $log_{10}(\text{MSE})(\downarrow)$ and std averaged on 4 random weight initializations on Graph Property Prediction tasks. The lower, the better. **First**, **second**, and **third** best results for each task are color-coded; we consider only the best configuration of GRAMA for coloring purposes.

| Model | Diameter | SSSP | Eccentricity |
|---|---|---|---|
| **MPNNs** | | | |
| GatedGCN | $0.1348_{\pm0.0397}$ | $-3.2610_{\pm0.0514}$ | $0.6995_{\pm0.0302}$ |
| GCN | $0.7424_{\pm0.0466}$ | $0.9499_{\pm0.0001}$ | $0.8468_{\pm0.0028}$ |
| GAT | $0.8221_{\pm0.0752}$ | $0.6951_{\pm0.1499}$ | $0.7909_{\pm0.0222}$ |
| GraphSAGE | $0.8645_{\pm0.0401}$ | $0.2863_{\pm0.1843}$ | $0.7863_{\pm0.0207}$ |
| GIN | $0.6131_{\pm0.0990}$ | $-0.5408_{\pm0.4193}$ | $0.9504_{\pm0.0007}$ |
| GCNII | $0.5287_{\pm0.0570}$ | $-1.1329_{\pm0.0135}$ | $0.7640_{\pm0.0355}$ |
| ARMA | $0.7819_{\pm0.4729}$ | $0.0432_{\pm0.0981}$ | $\mathbf{0.2605}_{\pm0.0610}$ |
| **DE-GNNs** | | | |
| DGC | $0.6028_{\pm0.0050}$ | $-0.1483_{\pm0.0231}$ | $0.8261_{\pm0.0032}$ |
| GRAND | $0.6715_{\pm0.0490}$ | $-0.0942_{\pm0.3897}$ | $0.6602_{\pm0.1393}$ |
| GraphCON | $0.0964_{\pm0.0620}$ | $-1.3836_{\pm0.0092}$ | $0.6833_{\pm0.0074}$ |
| A-DGN | $\mathbf{-0.5188}_{\pm0.1812}$ | $-3.2417_{\pm0.0751}$ | $0.4296_{\pm0.1003}$ |
| SWAN | $\mathbf{-0.5981}_{\pm0.1145}$ | $\mathbf{-3.5425}_{\pm0.0830}$ | $\mathbf{-0.0739}_{\pm0.2190}$ |
| **Graph Transformers** | | | |
| GPS | $-0.5121_{\pm0.0426}$ | $\mathbf{-3.5990}_{\pm0.1949}$ | $0.6077_{\pm0.0282}$ |
| **Our** | | | |
| GRAMA$_{\text{GCN}}$ | $0.2577_{\pm0.0368}$ | $0.0095_{\pm0.0877}$ | $0.6193_{\pm0.0441}$ |
| GRAMA$_{\text{GATEDGCN}}$ | $-0.5485_{\pm0.1489}$ | $\mathbf{-4.1289}_{\pm0.0988}$ | $0.5523_{\pm0.0511}$ |
| GRAMA$_{\text{GPS}}$ | $\mathbf{-0.8663}_{\pm0.0514}$ | $-3.9349_{\pm0.0699}$ | $\mathbf{-1.3012}_{\pm0.1258}$ |

works. This demonstrates that our method can effectively leverage the strengths of simple models while overcoming their limitations in long-range propagation.

### 5.3. Long-Range Benchmark

**Setup.** We assess the performance of our method on the real-world long-range graph benchmark (LRGB) from (Dwivedi et al., 2022b), focusing on the *Peptides-func* and *Peptides-struct* datasets. We follow the experimental setting in (Dwivedi et al., 2022b), including the 500K parameter budget. All transformer baselines include Laplacian positional encodings, for a fair evaluation. Our GRAMA does not use additional encodings. The datasets consist of large molecular graphs derived from peptides, where the structure and function of a peptide depend on interactions between distant parts of the graph. Therefore, relying on short-range interactions, such as those captured by local message passing in GNNs, may not be insufficient to excel at this task. More

Table 2: Results for Peptides-func and Peptides-struct averaged over 3 training seeds. Baselines are taken from (Dwivedi et al., 2022b) and (Gutteridge et al., 2023). All MPNN-based methods include structural and positional encoding. The first, second, and third best scores are colored, and we color only the best configuration of GRAMA.

| Model | Peptides-func AP ↑ | Peptides-struct MAE ↓ |
|---|---|---|
| **MPNNs** | | |
| GCN | $59.30_{\pm 0.23}$ | $0.3496_{\pm 0.0013}$ |
| GatedGCN | $58.64_{\pm 0.77}$ | $0.3420_{\pm 0.0013}$ |
| ARMA | $64.08_{\pm 0.62}$ | $0.2709_{\pm 0.0016}$ |
| **Multi-hop GNNs** | | |
| DIGL+MPNN+LapPE | $68.30_{\pm 0.26}$ | $0.2616_{\pm 0.0018}$ |
| MixHop-GCN+LapPE | $68.43_{\pm 0.49}$ | $0.2614_{\pm 0.0023}$ |
| DRew-GCN+LapPE | $71.50_{\pm 0.44}$ | $0.2536_{\pm 0.0015}$ |
| DRew-GatedGCN+LapPE | $69.77_{\pm 0.26}$ | $0.2539_{\pm 0.0007}$ |
| **Graph Transformers** | | |
| Transformer+LapPE | $63.26_{\pm 1.26}$ | $0.2529_{\pm 0.0016}$ |
| SAN+LapPE | $63.84_{\pm 1.21}$ | $0.2683_{\pm 0.0043}$ |
| GraphGPS+LapPE | $65.35_{\pm 0.41}$ | $0.2500_{\pm 0.0005}$ |
| **DE-GNNs** | | |
| GRAND | $57.89_{\pm 0.62}$ | $0.3418_{\pm 0.0015}$ |
| GraphCON | $60.22_{\pm 0.68}$ | $0.2778_{\pm 0.0018}$ |
| A-DGN | $59.75_{\pm 0.44}$ | $0.2874_{\pm 0.0021}$ |
| SWAN | $67.51_{\pm 0.39}$ | $0.2485_{\pm 0.0009}$ |
| **Graph SSMs** | | |
| Graph-Mamba | $67.39_{\pm 0.87}$ | $0.2478_{\pm 0.0016}$ |
| GMN | $70.71_{\pm 0.83}$ | $0.2473_{\pm 0.0025}$ |
| **Ours** | | |
| GRAMA$_{\text{GCN}}$ | $70.93_{\pm 0.78}$ | $0.2439_{\pm 0.0017}$ |
| GRAMA$_{\text{GatedGCN}}$ | $70.49_{\pm 0.51}$ | $0.2459_{\pm 0.0020}$ |
| GRAMA$_{\text{GPS}}$ | $69.83_{\pm 0.83}$ | $0.2436_{\pm 0.0022}$ |

details on the setup and tasks can be found in Appendix D.4.

**Results.** Table 2 provides a comparison of our GRAMA model with a wide range of baselines. A broader comparison is presented in Table 8. The results indicate that GRAMA outperforms standard MPNNs, transformer-based GNNs, DE-GNNs, SSM-based GNNs, and most Multi-hop GNNs. Such a result highlights the competitiveness of our method and its ability to propagate information effectively. Moreover, its empirical advantage over existing Graph SSMs emphasizes the strength of GRAMA in modeling long-range interactions while maintaining permutation equivariance and processing sequences that go beyond pairwise interactions. Similarly to Section 5.2, our results show that GRAMA strengthens the abilities of simple GNN backbones. Specifically, our method boosts GCN and GatedGCN by more than 11 AP points on the Peptide-func task.

## 6. Conclusion

We introduced GRAMA, a novel sequence-based framework that enhances the long-range interaction modeling ability and feature update selectivity of Graph Neural Networks (GNNs) through the integration of adaptive neural Autoregressive Moving Average (ARMA) models with potentially any GNN backbone. We draw a theoretical link between SSM models and GRAMA, to build solid groundwork and understanding of the qualitative behavior of GRAMA. Compared with several existing Graph SSMs, our GRAMA allows to benefit from long-range interaction modeling abilities, while maintaining permutation equivariance. Through a series of extensive experiments on 26 synthetic and real-world datasets, we demonstrated that GRAMA consistently offers competitive performance with well-established baseline models, from classical MPNNs to more complex approaches such as Graph Transformers and Graph SSMs. Overall, GRAMA offers a theoretically grounded, powerful, and flexible solution that bridges the gap between contemporary sequential models and existing graph learning methods, stepping forward towards a new family of graph machine learning models.

## Impact Statement

This paper aims to contribute to the field of Machine Learning, specifically focusing on advancing Graph Neural Networks (GNNs). It proposes a theoretically grounded and flexible solution that bridges the gap between contemporary sequential models and existing graph learning methods, paving the way for a new family of graph machine learning models. The research presented herein has a positive impact on the ongoing exploration and applications of GNNs coupled with State Space Models (SSMs).

In this work, we do not release any datasets or models that could pose a significant risk of misuse. We believe our research does not have any direct or indirect negative societal implications or harmful consequences, as we do not utilize sensitive, privacy-related data, nor do we develop methods that could be applied for harmful purposes. As far as we are aware, this study does not raise any ethical concerns or potential negative impacts. Furthermore, our research does not involve human subjects, nor does it employ crowdsourcing methods. We confirm there are no potential conflicts of interest or sponsorship influences affecting the objectivity or outcomes of this study.

## Acknowledgements

The work has been partially supported by EU-EIC EMERGE (Grant No. 101070918), and by NEURONE, a project funded by the European Union - Next Generation EU, M4C1 CUP I53D23003600006, under program PRIN 2022 (prj code 20229JRTZA). M.E. is funded by the Blavatnik-Cambridge fellowship, the Cambridge Accelerate Programme for Scientific Discovery, and the Maths4DL EPSRC Programme.

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

## Author Contributions

AG and AC recognized the problem of developing a principled integration of State Space Models in Graph Neural Networks. ME proposed the connection between SSMs and ARMA models, and conceptualized the methodology and method development. ME and AG conducted the empirical evaluation. The theoretical analysis was carried out by ME and AC. The original draft was written by ME, AG, and AC. DB, CG, and CBS contributed to the review and editing of the manuscript and supervised the project.

## A. Additional Related Work

**GNNs based on Differential Equations.** Building on the interpretation of convolutional neural networks (CNNs) as discretizations of ODEs and PDEs (Ruthotto & Haber, 2020; Chen et al., 2018; Zhang et al., 2019), several works, including GCDE (Poli et al., 2019), GODE (Zhuang et al., 2020), and GRAND (Chamberlain et al., 2021), among others, view GNN layers as discretized steps of the heat equation. This framework helps manage diffusion (smoothing) and sheds light on the oversmoothing problem in GNNs (Nt & Maehara, 2019; Oono & Suzuki, 2020; Cai & Wang, 2020). In contrast, Choromanski et al. (2022) introduced an attention mechanism based on the heat diffusion kernel. Other models, such as PDE-GCN$_M$ (Eliasof et al., 2021) and GraphCON (Rusch et al., 2022), combine diffusion with oscillatory processes to maintain feature energy. Recent work has explored anti-symmetry (Gravina et al., 2023; 2024a; 2025), reaction-diffusion dynamics (Wang et al., 2023; Choi et al., 2023; Eliasof et al., 2024b), convection (Zhao et al., 2023), advection (Eliasof et al., 2023b), port-Hamiltonian systems (Heilig et al., 2025), fractional Laplacian ODEs (Maskey et al., 2023), and higher-order methods (Eliasof et al., 2024c; Kang et al., 2024). While most of the aforementioned methods works in the setting of static graphs, temporal aspects are addressed in (Gravina et al., 2024b;c). Overall, we refer to this family of models as DE-GNNs. These models are related to SSM models, which are also based on ODEs. Also, some of the DE-GNNs were shown to be effective against oversquashing as architectures, and therefore we include them in our experimental comparisons.

**Multi-hop GNNs.** Multi-hop GNN architectures were extensively studied in previous years, leading to several popular architectures such as JK-Net (Xu et al., 2018), MixHop (Abu-El-Haija et al., 2019), and more recently DRew (Gutteridge et al., 2023). These works take inspiration from earlier works like DenseNets (Huang et al., 2017), where the main idea is to consider a combination of feature maps from multiple layers, instead of only considering the last layer feature map as in ResNets (He et al., 2016). We now distinguish our GRAMA from JK-Net, MixHop, and DRew. First, these methods do not stem from a dynamical system perspective that allows the construction of models like ARMA or SSM. Second, methods like JK-Net can become computationally expensive if many layers are used within a network, as it considers all previous layers, and it is only used within the final layer in a GNN, rather than an architecture that considers multiple past values at each layer of the network. Third, in GRAMA we propose a selective attention mechanism to ARMA coefficients, as described in Section 3.2. Compared with MixHop, which performs dense, non-recurrent projections, GRAMA uses a recurrent, non-dense aggregation inspired by dynamical systems and modern RNNs. A deeper theoretical and empirical comparison with MixHop is discussed in Appendix E.5.

**Transforming Graphs to Sequences.** In recent years, there has been growing interest in transforming graphs into sequences, with a substantial body of work addressing this problem. The motivation behind this transformation is to leverage well-established sequential learning mechanisms, such as 1D convolutions (Niepert et al., 2016; Eliasof et al., 2022; Sun et al., 2023), RNNs (Murphy et al., 2019a; Huang et al., 2022), and GRUs and LSTMs (Murphy et al., 2019b). Overall, these works propose various methods for converting graphs into sequences, often accompanied by theoretical insights into permutation equivariance, typically achieved in expectation under such transformations. In contrast, our GRAMA, adopts a different perspective: rather than mapping a graph to a single sequence, we construct a sequence of graphs. This design preserves permutation equivariance and enables the modeling of long-range interactions through our neural ARMA framework.

**Adaptivity in Graph Learning.** In recent years, it has been increasingly recognized that adaptivity plays a crucial role in graph learning, both in improving downstream performance and addressing fundamental limitations of classical GNNs, such as oversmoothing and oversquashing. Notably, adaptive mechanisms have proven effective in graph normalization (Eliasof et al., 2024a) and activation layers (Mantri et al., 2024), enhancing theoretical expressiveness and functional flexibility, respectively. Adaptivity has also been applied to message-passing schemes by enabling selective updates to node features, as explored from various perspectives in Sun et al. (2024); Errica et al. (2024). In this work, we take a different approach by introducing selectivity through learned ARMA coefficients. Specifically, our method, GRAMA, adaptively weighs previous states and residuals based on the input, allowing the model to dynamically modulate states and residuals dependencies.

**Expressiveness of GNNs.** The expressiveness of graph neural networks (GNNs) is a central aspect of graph representation learning, with much of the theoretical understanding framed through the Weisfeiler-Lehman (WL) test. Seminal works by Xu et al. (2019); Morris et al. (2019) established that the representational capacity of standard message-passing neural networks (MPNNs) is bounded by the power of the 1-WL test. This insight has motivated a broad line of research aimed at overcoming these limitations, including the use of random node initializations (Sato et al., 2021; Abboud et al., 2021), positional encodings (Eliasof et al., 2023a; Huang et al., 2024a), and subgraph-based architectures (Bevilacqua et al., 2022; 2024), among other approaches and methods. Within this theoretical landscape, our GRAMA is constructed to at least match the expressiveness of its underlying GNN backbone. This is achieved through its recurrent structure, which updates representations by aggregating both current and past hidden states along with residual correction terms. When the recurrence is simplified to use only the current state $\mathbf{f}^{(\ell)}$ while discarding residuals $\boldsymbol{\delta}^{(\tilde{\ell})}$, $\tilde{\ell} \leq \ell$, the resulting computation reduces to the forward pass of the backbone GNN, thereby preserving its expressiveness. Importantly, GRAMA allows more diverse behaviors — by learning data-dependent combinations of historical states and residuals, it expands the space of representable functions. Empirical results demonstrate that this enhanced flexibility consistently leads to improved performance over backbone models. Understanding if, and to which extent, our GRAMA improves expressiveness beyond 1-WL, presents a promising avenue for future theoretical investigation.

**Distinguishing GRAMA from other GNNs.** We now further distinguish our GRAMA from other types of existing GNNs. compared with other graph SSMs, the main differences are (i) GRAMA can process sequences which are beyond pairwise interactions, different from the graph SSM in (Huang et al., 2024b); and (ii) GRAMA is permutation-equivariant, while other graph SSMs like in (Behrouz & Hashemi, 2024; Wang et al., 2024a), are not permutation-equivariant and are based on heuristics that order the graph nodes to obtain a sequence. Compared to Ding et al. (2024), which uses an LRU-based mechanism without selective control, GRAMA incorporates a selective mechanism, which our results in Tables 9, 11 and 12 show to be impactful. Compared with transformers like GRIT (Ma et al., 2023), we differ in both computational cost and operation. GRIT emphasizes expressive positional encodings and is more resource-intensive than GPS, whereas GRAMA is efficient, permutation-equivariant, and designed for long-range propagation. Compared with implicit GNNs such as IGNN (Gu et al., 2020) and GIND (Chen et al., 2022), which model graph representations as fixed points of nonlinear equilibrium equations over static graphs, leveraging global aggregation through learned diffusion or optimization-based formulations, our GRAMA instead views graph modeling by constructing a sequence of graphs. This enables GRAMA to perform explicit, state- and residual-dependent aggregation via learned adaptive ARMA dynamics, which are also equivalent to SSMs, as we prove in Appendix B.

## B. Proofs

We now provide proof to all Theorems and Lemmas shown in the paper. Without loss of generality, we analyze GRAMA in the case of a single channel. However, note that the ARMA coefficients are shared among channels. Therefore, in the case of multiple input channels, the proof is trivially extended by applying the same ARMA system to each channel independently. Moreover, we will state our theoretical results considering general sequences indexed with $t$, which in particular can be thought of as neural sequences of GRAMA, but, for the sake of simplicity, without involving the hyperparameters $R$ and $S$, and focusing on the dynamics of a single GRAMA block.

### B.1. Proof of Theorem 4.1

*Proof.* We start by recapping the definition of the ARMA and linear SSM models. Then, we show how to derive an SSM representation of an ARMA model, and vice versa, an ARMA model from a linear SSM.

**ARMA Models.** The ARMA$(p, q)$ model for a univariate time series is given by:

$$f_t = \phi_1 f_{t-1} + \phi_2 f_{t-2} + \ldots + \phi_p f_{t-p} + \delta_t + \theta_1 \delta_{t-1} + \theta_2 \delta_{t-2} + \ldots + \theta_q \delta_{t-q}, \tag{9}$$

where $\{\phi_i\}_{i=1}^p$ are the autoregressive coefficients, and $\{\theta_j\}_{j=1}^q$ are the moving average coefficients.

**State Space Model (SSM):** A linear SSM system mapping univariate input, $\delta_t$, into univariate output, $f_t$, is defined by the following equations:

$$\mathbf{x}_t = \mathbf{A}\mathbf{x}_{t-1} + \mathbf{B}\delta_t. \tag{10a}$$

$$f_t = \mathbf{C}\mathbf{x}_t + \mathbf{D}\delta_t, \tag{10b}$$

where $\mathbf{x}_t$ is the hidden state vector at time step $t$, $\mathbf{A}$ is the state transition matrix, $\mathbf{B}$ is the control-input matrix, $\mathbf{C}$ is the observation matrix, and $\mathbf{D}$ is the direct transition matrix.

**Proof of ARMA $\rightarrow$ SSM.** Given an ARMA$(p,q)$ model, we can rewrite it in an SSM form by defining a state vector $\mathbf{x}_t$ that includes past autoregressive values and past residuals:

$$\mathbf{x}_t = \begin{bmatrix} f_t & f_{t-1} & \ldots & f_{t-p+1} & \delta_t & \delta_{t-1} & \ldots & \delta_{t-q+1} \end{bmatrix}^\top \tag{11}$$

and define the SSM matrices $\mathbf{A}, \mathbf{B}, \mathbf{C}, \mathbf{D}$, as follows:

$$\mathbf{A} = \left[ \begin{array}{ccccc|ccccc} \phi_1 & \phi_2 & \ldots & \phi_{p-1} & \phi_p & \theta_1 & \theta_2 & \ldots & \theta_{q-1} & \theta_q \\ 1 & 0 & \ldots & 0 & 0 & 0 & 0 & \ldots & 0 & 0 \\ 0 & 1 & \ldots & 0 & 0 & 0 & 0 & \ldots & 0 & 0 \\ \vdots & \vdots & \ddots & \vdots & \vdots & \vdots & \vdots & \ddots & \vdots & \vdots \\ 0 & 0 & \ldots & 1 & 0 & 0 & 0 & \ldots & 0 & 0 \\ \hline 0 & 0 & \ldots & 0 & 0 & 0 & 0 & \ldots & 0 & 0 \\ 0 & 0 & \ldots & 0 & 0 & 1 & 0 & \ldots & 0 & 0 \\ 0 & 0 & \ldots & 0 & 0 & 0 & 1 & \ldots & 0 & 0 \\ \vdots & \vdots & \ddots & \vdots & \vdots & \vdots & \vdots & \ddots & \vdots & \vdots \\ 0 & 0 & \ldots & 0 & 0 & 0 & 0 & \ldots & 1 & 0 \end{array} \right] \tag{12}$$

$$\mathbf{B} = \begin{bmatrix} 0 & \ldots & 0 & | & 1 & 0 & \ldots & 0 \end{bmatrix}^\top \tag{13}$$

$$\mathbf{C} = \begin{bmatrix} 1 & 0 & \ldots & 0 & 0 \end{bmatrix}, \quad \mathbf{D} = \begin{bmatrix} 1 \end{bmatrix} \tag{14}$$

Using these definitions, the obtained state space model representation is equivalent to the operation of the ARMA model of Equation (9).

**SSM $\rightarrow$ ARMA.** Let us assume the hidden state dimension to be $p$, so that $\mathbf{A} \in \mathbb{R}^{p \times p}, \mathbf{B} \in \mathbb{R}^{p \times 1}, \mathbf{C} \in \mathbb{R}^{1 \times p}, \mathbf{D} \in \mathbb{R}^{1 \times 1}$. First, we recursively substitute the state equation into itself to express $\mathbf{x}_t$ in terms of past states and inputs. Substituting $\mathbf{x}_{t-1}$ into the Equation (10) yields:

$$\mathbf{x}_t = \mathbf{A}\mathbf{x}_{t-1} + \mathbf{B}\delta_t = \mathbf{A}(\mathbf{A}\mathbf{x}_{t-2} + \mathbf{B}\delta_{t-1}) + \mathbf{B}\delta_t \tag{15}$$
$$= \mathbf{A}^2\mathbf{x}_{t-2} + \mathbf{A}\mathbf{B}\delta_{t-1} + \mathbf{B}\delta_t$$

Therefore, unfolding $t$ steps in the past, up to the initial condition $\mathbf{x}_0$, we get:

$$\mathbf{x}_t = \mathbf{A}^t\mathbf{x}_0 + \sum_{k=0}^{t-1} \mathbf{A}^k\mathbf{B}\delta_{t-k} = \mathbf{A}^t\mathbf{x}_0 + \mathbf{B}\delta_t + \sum_{k=1}^{t-1} \mathbf{A}^k\mathbf{B}\delta_{t-k} \tag{16}$$

Substituting the expression in Equation (16) to obtain the SSM output from Equation (10b) yields:

$$f_t = \mathbf{C}\left( \mathbf{A}^t\mathbf{x}_0 + \mathbf{B}\delta_t + \sum_{k=1}^{t-1} \mathbf{A}^k\mathbf{B}\delta_{t-k} \right) + \mathbf{D}\delta_t \tag{17}$$

$$= \mathbf{C}\mathbf{A}^t\mathbf{x}_0 + (\mathbf{C}\mathbf{B} + \mathbf{D})\delta_t + \sum_{k=1}^{t-1} \mathbf{C}\mathbf{A}^k\mathbf{B}\delta_{t-k}$$

The above equation describes an ARMA(p,q) model, where $p = t$, and $q = p - 1$. In fact, once defined the initial condition as $\mathbf{x}_0 = [f_{p-1}, \ldots, f_0]^T$, the autoregressive coefficients can be found as the $p$ elements of the row vector $\mathbf{C}\mathbf{A}^p \in \mathbb{R}^{1 \times p}$. While, the moving average coefficients are the $q$ real numbers defined as $\theta_k = \mathbf{C}\mathbf{A}^k\mathbf{B}$, for $k = 1, \ldots, q$. Finally, to get exactly the form of Equation (9), it suffices to impose that $\mathbf{D} = 1 - \mathbf{C}\mathbf{B}$. $\qquad\square$

Another proof of the equivalence between ARMA and SSM can be found in (de Jong & Penzer, 2004). We developed our own version since it is more congenial to our discussion based on long-term propagation of the information on graphs.

## B.2. Proof of Lemma 4.2

The linear SSM corresponding to a GRAMA block with autoregressive coefficients $\{\phi_i\}_{i=1}^p$ is stable if and only if the spectral radius of its state matrix is less than (or at most equal to) 1. In particular, this happens if and only if the polynomial $P(\lambda) = \lambda^p - \sum_{j=1}^p \phi_j \lambda^{p-j}$ has all its roots inside (or at most on) the unit circle.

*Proof.* We proved in Theorem 4.1 that a GRAMA block can be described equivalently as a linear SSM of the kind of Equation (10). The discrete-time recurrence given by Equation (10) can be completely unfolded, thanks to the lack of nonlinearity. We can write Equation (10) in a closed formulation as

$$\mathbf{x}_t = \mathbf{A}^t \mathbf{x}_0 + \sum_{j=0}^{t-1} \mathbf{A}^j \mathbf{B} \delta_{t-j}. \tag{18}$$

A necessary and sufficient condition to have a bounded response for the state $\mathbf{x}_t$ is that the powers of the state matrix $\mathbf{A}$ do not explode. This condition translates into a well-known inequality on the spectral radius of the state matrix, namely that the spectral radius of $\mathbf{A}$ is less than (or at most equal to) 1.

Now, let us consider the state matrix as in Equation (12), i.e. divided in an upper triangular form of 4 blocks: $\mathbf{A}_{11}, \mathbf{A}_{12}, \mathbf{A}_{21}, \mathbf{A}_{22}$ of dimensions $p \times p, p \times q, q \times p, q \times q$, where $\mathbf{A}_{21}$ is the null matrix of dimension $q \times p$. Due to the triangular form, we have that $\det(\mathbf{A} - \lambda \mathbf{I}) = \det(\mathbf{A}_{11} - \lambda \mathbf{I}) \det(\mathbf{A}_{22} - \lambda \mathbf{I}) = \det(\mathbf{A}_{11} - \lambda \mathbf{I})(-1)^q \lambda^q$. The matrix $\mathbf{A}_{11}$ is a companion matrix. Its characteristic polynomial can be computed recursively using Laplace expansion of determinants on the first row, to get that $\det(\mathbf{A}_{11} - \lambda \mathbf{I}) = (-1)^p \left( \lambda^p - \sum_{j=1}^p \phi_j \lambda^{p-j} \right)$. Therefore, the set of eigenvalues of the state matrix of the linear SSM associated with a GRAMA block with autoregressive coefficients $\{\phi_i\}_{i=1}^p$ is the set of roots of the polynomial in the indeterminate $\lambda$, given by

$$(-1)^{p+q} \lambda^q \left( \lambda^p - \sum_{j=1}^p \phi_j \lambda^{p-j} \right).$$

The spectral radius of $\mathbf{A}$ is the largest (in modulo) among all the complex roots of this polynomial. Thus, a GRAMA block with autoregressive coefficients $\{\phi_i\}_{i=1}^p$ is stable if and only if the polynomial $P(\lambda) = \lambda^p - \sum_{j=1}^p \phi_j \lambda^{p-j}$ has all its roots inside (or at most on) the unit circle. $\square$

## B.3. Proof of Theorem 4.3

If $\sum_{j=1}^p |\phi_j| \le 1$, then the GRAMA block with autoregressive coefficients $\{\phi_i\}_{i=1}^p$ corresponds to a stable linear SSM.

*Proof.* Consider the polynomial $P(\lambda) = \lambda^p - \sum_{j=1}^p \phi_j \lambda^{p-j}$. The Lagrange upper bound (Hirst & Macey, 1997, Theorem 1) states that all the complex roots of $P(\lambda)$ have modulus less or equal than $\max\{1, \sum_{j=1}^p |\phi_j|\}$. Therefore, if $\sum_{j=1}^p |\phi_j| \le 1$ then, from Lemma 4.2, we conclude that the linear SSM corresponding to our GRAMA block with autoregressive coefficients $\{\phi_i\}_{i=1}^p$ is stable. $\square$

## B.4. Proof of Theorem 4.4

First, we prove the following Lemma B.1.

**Lemma B.1** (Long-range interactions in GRAMA). *If the $k$-th power of the state matrix of a GRAMA block has vanishing entries, then for a stable GRAMA it is impossible to learn long-term dependencies of $k$ time lags in the sequence of residuals $\delta_1, \delta_2, \ldots, \delta_t$.*

*Proof.* Assuming we want to learn patterns in the input sequence $\delta_1, \delta_2, \ldots, \delta_t$ of length $k$. Referring to Equation (18), we need the current hidden state $\mathbf{x}_t$ to encode information that was present in $\delta_{t-k}$. Now, if $\mathbf{A}^k$ has vanishing entries, i.e., smaller than machine precision, then the same holds for the vector $\mathbf{A}^k \mathbf{B}$. Ergo, it is impossible to implement a linear SSM, or equivalently an ARMA model, to learn dependencies in the input of length $k$. $\square$

Now, we can prove Theorem 4.4, whose statement we report here below for ease of comprehension.
Let us be given a GRAMA block with autoregressive coefficients $\{\phi_i\}_{i=1}^p$. Assume the roots of the polynomial

$P(\lambda) = \lambda^p - \sum_{j=1}^{p} \phi_j \lambda^{p-j}$ are all inside the unit circle. Then, the closer the roots $P(\lambda)$ are to the unit circle, the longer the range propagation of the linear SSM corresponding to such a GRAMA block.

*Proof.* Due to Lemma 4.2, the hypothesis of $P(\lambda)$ having roots inside the unit circle implies that the linear SSM corresponding to a GRAMA block with autoregressive coefficients $\{\phi_i\}_{i=1}^{p}$ is a stable system. Moreover, from the proof of Lemma B.1, we know that the long-term propagation of a stable linear SSM corresponding to a GRAMA block is prevented by the pace to which the vector $\mathbf{A}^k \mathbf{B}$ converges to the zero vector, as $k$ increases. The speed of convergence is linked to the speed of convergence of $\mathbf{A}^k$ to the null matrix, which in turn depends on the modulus of the eigenvalues of $\mathbf{A}$. From Lemma 4.2, the non-zero eigenvalues of $\mathbf{A}$ are the roots of the polynomial $P(\lambda)$. Therefore, the closer the moduli of the complex roots of the polynomial $P(\lambda)$ are to the unit circle, the longer the range propagation of the GRAMA block. $\square$

## C. Implementation Details

We provide additional implementation details of our GRAMA.

### C.1. Learning Selective ARMA Coefficients

We now describe the implementation of the Selective ARMA coefficients presented in Section 3.2. Namely, to learn the dynamics between node features in different steps within the sequences $\mathbf{L}^{(s)}$ and $\mathbf{\Delta}^{(s)}$, we utilize a multi-head self-attention mechanism (Vaswani et al., 2017). Recall that the shape of the sequences is $L \times n \times d$, where $L$ is the sequence length, $n$ is the number of nodes, and $d$ is the number of hidden channels. To maintain computational efficiency, we first mean pool the sequences along the node dimension (per graph), such that the input to the attention layers is of shape $L \times c$. We denote this operation by POOL, and it is a common operation in graph learning (Xu et al., 2019; Morris et al., 2019). This pooling step allows our GRAMA to offer flexible behavior in terms of ARMA coefficients per graph, a property which was recently shown to be effective in graph learning (Eliasof et al., 2024a; Mantri et al., 2024) while remaining efficient in terms of computations. In what follows, we explain how to obtain the ARMA coefficients using an attention mechanism. For simplicity, we describe the process in the case of $p = q = L$. In any other case, the exact computation is done with a truncated version of the sequence, taking the latest $p$ sequence elements from $\mathbf{F}^{(s)} \in \mathbb{R}^{L \times n \times d}$ (in Python notations, $\mathbf{F}^{(s)}[:-p,:,:]$, and the last $q$ sequence elements from $\mathbf{\Delta}^{(s)}$ (in Python notations, $\mathbf{\Delta}^{(s)}[:-q,:,:]$) That is, the truncated are fed to the attention layers as described below. In terms of using an attention mechanism, the main difference in our implementation compared to a standard attention module as in Vaswani et al. (2017) is that we remove the SoftMax normalization step, as discussed in Section 3.2. We denote a multi-head attention score module by $\mathrm{MHA_{AR}}$ and $\mathrm{MHA_{MA}}$, for the multi-head-attention for the AR and MA parts, respectively. Note that in our case, we are only interested in the pairwise scores computed within a transformer and that we do not use the SoftMax normalization step. Then, the output of the attention modules reads:

$$\mathcal{A}_{\mathbf{F}^{(s)}} = tanh\left(\mathrm{MHA_{AR}}(\mathrm{POOL}(\mathbf{F}^{(s)}))\right) \in \mathbb{R}^{L \times L}, \tag{19}$$

$$\mathcal{A}_{\mathbf{\Delta}^{(s)}} = tanh\left(\mathrm{MHA_{MA}}(\mathrm{POOL}(\mathbf{\Delta}^{(s)}))\right) \in \mathbb{R}^{L \times L}. \tag{20}$$

The $(l_i, l_j)$-th entries in $\mathcal{A}_{\mathbf{F}^{(s)}}$ and $\mathcal{A}_{\mathbf{\Delta}^{(s)}}$ represent the score between the $l_i$-th and $l_j$-th elements in the sequences, respectively. Specifically, the last row of these matrices represents the connection between the current element $l$ and the elements $L-1$ in each of the respective sequences. Therefore, we define the unnormalized AR coefficients as the last row in $\mathcal{A}_{\mathbf{F}^{(s)}}$, and similarly in $\mathcal{A}_{\mathbf{\Delta}^{(s)}}$ for the MA coefficients. Using Python notations, this is described as:

$$\tilde{\mathbf{c}}_{\mathrm{AR}}(\mathbf{F}^{(s)}) = \mathcal{A}_{\mathbf{F}^{(s)}}[-1,:] \in \mathbb{R}^{L}, \tag{21}$$

$$\tilde{\mathbf{c}}_{\mathrm{MA}}(\mathbf{\Delta}^{(s)}) = \mathcal{A}_{\mathbf{\Delta}^{(s)}}[-1,:] \in \mathbb{R}^{L}. \tag{22}$$

To normalize the coefficients, we follow the following strategy:

$$\mathbf{c}_{\mathrm{AR}}(\mathbf{F}^{(s)}) = \frac{\tilde{\mathbf{c}}_{\mathrm{AR}}(\mathbf{F}^{(s)})}{\sum \tilde{\mathbf{c}}_{\mathrm{AR}}(\mathbf{F}^{(s)})}, \tag{23}$$

$$\mathbf{c}_{\mathrm{MA}}(\mathbf{\Delta}^{(s)}) = \frac{\tilde{\mathbf{c}}_{\mathrm{MA}}(\mathbf{\Delta}^{(s)})}{\sum \tilde{\mathbf{c}}_{\mathrm{MA}}(\mathbf{\Delta}^{(s)})}. \tag{24}$$

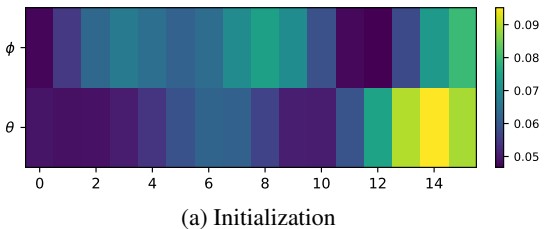 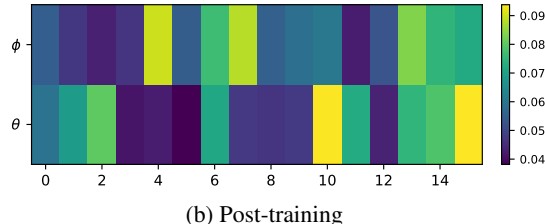

(a) Initialization                              (b) Post-training

Figure 3: Learned coefficients of GRAMA over layers, visualized before (a) and after (b) training.

In Figure 3, we present an example of the learned coefficients before and after training on the Questions dataset, using a single GRAMA block with 16 recurrence steps. At initialization, the weights assigned to layerwise features—denoted by $\phi$ for hidden states and $\theta$ for residuals—are predominantly concentrated on the deeper layers. This initialization resembles a standard residual connection, where emphasis is placed on recent computations. Following training, the learned coefficients reveal a more intricate distribution. Rather than attending primarily to the most recent layer, the model highlights a diverse set of layers, suggesting non-trivial dependencies.

### C.2. Overall GRAMA architecture

Our GRAMA is illustrated in Figure 1, and it is comprised of three main components: (i) the initial embedding, which is described in Equation (2). The role of this part is to transform a static input graph into a sequence of graph inputs. Namely, given features of shape $n \times c$, it yields two sequences: A sequence of states $\mathbf{F}^{(0)}$, and a sequence of residuals $\mathbf{\Delta}^{(0)}$, both of shape $L \times n \times d$, where $d$ is the embedding size of the input $c$ channels. (ii) These sequences are then processed by a GRAMA block, as discussed in Section 3.1. (iii) A final classifier $g_{\text{out}} : \mathbb{R}^d \rightarrow \mathbb{R}^o$ that takes the last state in the updated sequence, denoted by $\mathbf{f}^{(L \cdot S - 1)} \in \mathbb{R}^{n \times d}$, and projects it to the desired number of output channels. The classifier is implemented using an MLP, as is standard in graph learning (Xu et al., 2019). Note that, the last state $\mathbf{f}^{(L \cdot S - 1)}$ contains node features, and therefore, in the case of a graph level task, we first pool the node features using mean pooling as in (Xu et al., 2019), to obtain a prediction vector $g_{\text{out}}(\text{POOL}(\mathbf{f}^{(L \cdot S - 1)})) \in \mathbb{R}^o$. In the case of node-level tasks, the node-wise prediction is obtained by $g_{\text{out}}(\mathbf{f}^{(L \cdot S - 1)}) \in \mathbb{R}^{(n \times o)}$.

## D. Experimental Details

In this section, we provide additional experimental details.

**Compute.** Our experiments are run on NVIDIA A6000 and A100 GPUs, with 48GB and 80GB of memory, respectively.

### D.1. Employed baselines

In our experiments, the performance of our method is compared with various state-of-the-art GNN baselines from the literature. Specifically, we consider:

- classical MPNN-based methods, i.e., GCN (Kipf & Welling, 2016), GraphSAGE (Hamilton et al., 2017), GAT (Veličković et al., 2018), GatedGCN (Bresson & Laurent, 2018), GIN (Xu et al., 2019), ARMA (Bianchi et al., 2021), GINE (Hu et al., 2020b), GCNII (Chen et al., 2020), and CoGNN (Finkelshtein et al., 2024);

- heterophily-specific models, i.e., H2GCN (Zhu et al., 2020), CPGNN (Zhu et al., 2021), FAGCN (Bo et al., 2021), GPR-GNN (Chien et al., 2021), FSGNN (Maurya et al., 2022), GloGNN (Li et al., 2022), GBK-GNN (Du et al., 2022), and JacobiConv (Wang & Zhang, 2022);

- DE-DGNs, i.e., DGC (Wang et al., 2021), GRAND (Chamberlain et al., 2021), GraphCON (Rusch et al., 2022), A-DGN (Gravina et al., 2023), SWAN (Gravina et al., 2025), and PH-DGN (Heilig et al., 2025);

- Graph Transformers, i.e., Transformer (Vaswani et al., 2017; Dwivedi & Bresson, 2021), GT (Shi et al., 2021), SAN (Kreuzer et al., 2021a), EGT (Hussain et al., 2021), GPS (Rampášek et al., 2022), GOAT (Kong et al., 2023), Exphormer (Shirzad et al., 2023), GRIT (Ma et al., 2023), and Polynormer (Deng et al., 2024);

- Higher-Order DGNs, i.e., DIGL (Gasteiger et al., 2019), MixHop (Abu-El-Haija et al., 2019), DRew (Gutteridge et al., 2023), and GRED (Ding et al., 2024).

- SSM-based GNN, i.e., Graph-Mamba (Wang et al., 2024a), GMN (Behrouz & Hashemi, 2024), GSSM (Huang et al., 2024b), and GPS+Mamba (Behrouz & Hashemi, 2024)

### D.2. Graph Transfer

**Dataset.** We consider the graph transfer dataset from Gravina et al. (2025), which is based on the work of (Di Giovanni et al., 2023). Unlike the original approach in (Di Giovanni et al., 2023), node features are randomly sampled from a uniform distribution in the range $[0, 0.5]$. In each graph, labels of value "1" and "0" are assigned to a source node and a target node, respectively. Graphs were sampled from three different distributions: line, ring, and crossed-ring (see Figure 4 for a visual exemplification). In ring graphs, the nodes form a cycle of size $n$, with the source and target placed $\lfloor n/2 \rfloor$ apart. Similarly, crossed-ring graphs consisting of cycles of size $n$ but introduced additional edges crossing intermediate nodes, while still maintaining a source-target distance of $\lfloor n/2 \rfloor$. Lastly, the line graph contains a path of length $n$ between the source and target nodes. These experiments focus on a regression task aimed at swapping the labels of the source and target nodes while keeping intermediate node labels unchanged. The input dimension is 1, and the distances between source and target nodes are set to 3, 5, 10, and 50. We generated 1000 graphs for training, 100 for validation, and 100 for testing.

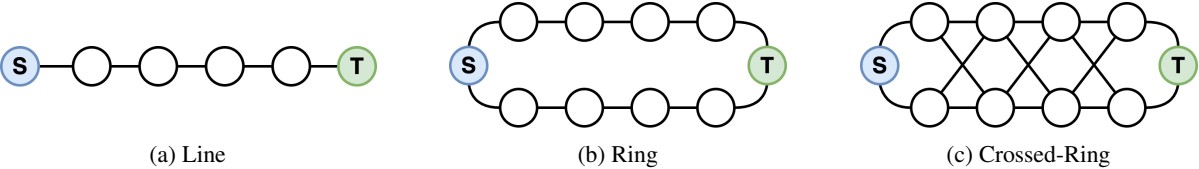

| (a) Line | (b) Ring | (c) Crossed-Ring |

Figure 4: Line, ring, and crossed-ring graphs where the distance between source and target nodes is equal to 5. Nodes marked with "S" are source nodes, while the nodes with a "T" are target nodes.

**Experimental Setting.** We followed the experimental setting of (Gravina et al., 2025). Therefore, we design each model as a combination of three main components. The first is the encoder which maps the node input features into a latent hidden space; the second is the graph convolution (i.e., GRAMA or the other baselines); and the third is a readout that maps the output of the convolution into the output space. The encoder and the readout share the same architecture among all models in the experiments.

We perform hyperparameter tuning via grid search, optimizing the Mean Squared Error (MSE) computed on the node features of the whole graph. We train the models using the Adam optimizer for a maximum of 2000 epochs and early stopping with a maximal patience of 100 epochs on the validation loss. For each model configuration, we perform 4 training runs with different weight initialization and report the average of the results. We report in Table 4 the grid of hyperparameters exploited for this experiment.

### D.3. Graph Property Prediction

**Dataset.** We adhered to the data generation procedure described in (Gravina et al., 2023). Graphs were randomly drawn from several distributions, e.g., Erdős–Rényi, Barabasi-Albert, caveman, tree, and grid. Each graph contains between 25 and 35 nodes, with nodes assigned with random identifiers as input features sampled from a uniform distribution in the range $[0, 1)$. The target values represent single-source shortest paths, node eccentricity, and graph diameter. The dataset included a total of 7,040 graphs, with 5,120 for training, 640 for validation, and 1,280 for testing. The tasks in this benchmark require capturing long-term dependencies between nodes, as solving them requires computing the shortest paths within the graph. Moreover, as described in Gravina et al. (2023), similar to standard algorithmic approaches (e.g., Bellman-Ford, Dijkstra's algorithm), accurate solutions depend on the exchange of multiple messages between nodes, making local information insufficient for this task. Additionally, the graph distributions used in these tasks are sampled from caveman, tree, line, star, caterpillar, and lobster distributions, all of which include bottlenecks by design, which are known to be a cause of oversquashing (Topping et al., 2022).

**Experimental Setting.** We employ the same datasets, hyperparameter space, and experimental setting presented in Gravina et al. (2023). Therefore, we perform hyperparameter tuning via grid search, optimizing the Mean Square Error (MSE),

training the models using Adam optimizer for a maximum of 1500 epochs, and early stopping with patience of 100 epochs on the validation error. For each model configuration, we perform 4 training runs with different weight initialization and report the average of the results. We report in Table 4 the grid of hyperparameters exploited for this experiment.

### D.4. Long Range Graph Benchmark

**Dataset.** To assess the performance on real-world long-range graph benchmarks, we considered the Peptides-func and Peptides-struct datasets (Dwivedi et al., 2022b). The graphs represent 1D amino acid chains, with nodes corresponding to the heavy (non-hydrogen) atoms of the peptides, and edges representing the bonds between them. *Peptides-func* is a multi-label graph classification dataset containing 10 classes based on peptide functions, such as antibacterial, antiviral, and cell-cell communication. *Peptides-struct* is a multi-label graph regression dataset, focused on predicting 3D structural properties of peptides. The regression tasks involve predicting the inertia of molecules based on atomic mass and valence, the maximum atom-pair distance, sphericity, and the average distance of all heavy atoms from the plane of best fit. Both datasets, Peptides-func and Peptides-struct, consist of 15,535 graphs, encompassing a total of 2.3 million nodes. We used the official splits from Dwivedi et al. (2022b), and reported the average and standard-deviation performance across 3 seeds.

**Experimental Setting.** We employ the same datasets and experimental setting presented in Dwivedi et al. (2022b). Therefore, we perform hyperparameter tuning via grid search, optimizing the Average Precision (AP) in the Peptide-func task and the Mean Absolute Error (MAE) in the Peptide-struct task, training the models using AdamW optimizer for a maximum of 300 epochs. For each model configuration, we perform 3 training runs with different weight initialization and report the average of the results. Also, we follow the guidelines in (Dwivedi et al., 2022b; Gutteridge et al., 2023) and stay within the 500K parameter budget. In Table 4 we report the grid of hyperparameters exploited for this experiment.

### D.5. GNN Benchmarks

**Dataset.** *MalNet-Tiny* (Freitas et al., 2021) is a graph classification dataset consisting of 5,000 function call graphs derived from software samples in the Android ecosystem. Each graph contains at most 5,000 nodes, which represent functions. Edges correspond to calls between functions. MalNet-Tiny is a graph classification dataset, comprising of 5 classification labels, including 1 benign software and 4 types of malware. We used stratified splitting, following a 70%-10%-20% split, as in Freitas et al. (2021).

In the heterophilic setting, we consider Roman-empire, Amazon-ratings, Minesweeper, Tolokers, and Questions tasks from (Platonov et al., 2023). *Roman-Empire* is a dataset derived from the Roman Empire article in Wikipedia. Each node represents a word, and edges are formed if words either follow one another or are connected syntactically. The task involves node classification based on the syntactic role of the word, with 18 classes. The graph is chain-like, has sparse connectivity, and potentially long-range dependencies. *Amazon-Ratings* is based on the Amazon product co-purchasing network. Nodes represent products, and edges connect products that are frequently bought together. The task is to predict the average product rating, which is grouped into five classes. Node features are derived from fastText embeddings of product descriptions. *Minesweeper* is a synthetic dataset consisting of a 100x100 grid where nodes represent cells, and edges connect neighboring cells. 20% of the nodes are randomly selected as mines. The task is to predict which nodes are mines, employing as node features the one-hot-encoded numbers of neighboring mines. *Tolokers* is a dataset based on the Toloka crowdsourcing platform (Likhobaba et al., 2023), where nodes represent workers (tolokers), and edges are formed if workers collaborate on the same project. The task is to predict whether a worker has been banned, using features from their profile and performance statistics. *Questions* is based on the data from the Yandex Q question-answering website. Nodes represent users, and edges connect users who have interacted by answering each other's questions. The task is to predict which users remained active on the platform, with node features derived from user descriptions. We report in Table 3 a summary of the statistics of the employed heterophilic datasets.

**Experimental Setting.** We employ the same datasets and experimental setting presented in Freitas et al. (2021) and (Platonov et al., 2023). Therefore, we perform hyperparameter tuning via grid search, optimizing the Accuracy (Acc) in the MalNet-Tiny, Roman-Empire, and Amazon-ratings tasks, and the ROC Area Under the Curve (AUC) in the Minesweeper, Tolokers, and Questions task. We trained the models using AdamW optimizer for a maximum of 300 epochs. On the heterophilic datasets, we use the official splits provided in Platonov et al. (2023) and report the average and standard deviation of the obtained performance. For MalNet-Tiny, we repeat the experiment on 4 different seeds and report the average performance alongside the standard deviation. We report in Table 4 the grid of hyperparameters considered for this experiment.

Table 3: Statistics of the heterophilous datasets.

|  | Roman-empire | Amazon-ratings | Minesweeper | Tolokers | Questions |
|---|---|---|---|---|---|
| N. nodes | 22,662 | 24,492 | 10,000 | 11,758 | 48,921 |
| N. edges | 32,927 | 93,050 | 39,402 | 519,000 | 153,540 |
| Avg degree | 2.91 | 7.60 | 7.88 | 88.28 | 6.28 |
| Diameter | 6,824 | 46 | 99 | 11 | 16 |
| Node features | 300 | 300 | 7 | 10 | 301 |
| Classes | 18 | 5 | 2 | 2 | 2 |
| Edge homophily | 0.05 | 0.38 | 0.68 | 0.59 | 0.84 |

## D.6. Hyperparameters

In Table 4, we report the grids of hyperparameters employed in our experiments by our method. Besides typical learning hyperparameters such as learning rate and weight decay, our GRAMA introduces several possible hypermeters: the sequence length $L$, the autoregressive order $p$, the moving average $q$, the number of recurrent steps applied at each GRAMA block, and the number of GRAMA blocks $S$. We now describe our choices, aiming to maintain a reasonable number of hyperparameters and to obtain a large prediction window, which was shown to be useful in SSMs (Gu et al., 2021c). Because this paper focuses on using a sequential model on static graph inputs, the sequence length is to be determined, and we consider different lengths depending on the task. The ARMA orders $p$ and $q$ can assume any values as long as they are not larger than $L$, and the most general case is when $p = q = L$, and this was our choice, as it covers other choices where $p$ or $q$ are smaller than $L$. For the number of recurrence steps $R$, we aim to obtain a relatively large prediction window with respect to the input sequence length, and therefore we choose to set $R = L$ in all experiments.

Table 4: The grid of hyperparameters employed during model selection for the graph transfer tasks (*Transfer*), graph property prediction tasks (*GraphProp*), Long Range Graph Benchmark (*LRGB*), and GNN benchmarks (*G-Bench*), i.e., MalNet-Tiny and heterophilic datasets.

| Hyperparameters | Values | | | |
|---|---|---|---|---|
|  | *Transfer* | *GraphProp* | *LRGB* | *G-Bench* |
| Optimizer | Adam | Adam | AdamW | AdamW |
| Learning rate | 0.001 | 0.003 | 0.001, 0.0005, 0.0001 | 0.001, 0.0005 ,0.0001 |
| Weight decay | 0 | $10^{-6}$ | 0, 0.0001 | 0, 0.0001 |
| Dropout | 0 | 0 | 0, 0.3, 0.5 | 0, 0.3, 0.5 |
| Activation function ($\sigma$) | ReLU | ReLU | ELU, GELU, ReLU | ELU, GELU, ReLU |
| Embedding dim (d) | 64 | 10, 20, 30 | 64, 128 | 64, 128, 256 |
| Sequence Length (L) | 1, 3, 5, 10, 50 | 1, 5, 10, 20 | 2, 4, 8, 16 | 2, 4, 8, 16 |
| Blocks (S) | 1, 2 | 1, 2 | 1, 2, 4 | 1, 2, 4 |
| Graph Backbone | GCN, GPS, GatedGCN | | | |

# E. Additional Results And Comparisons

## E.1. Additional GNN Benchmarks

**Setup.** To further evaluate the performance of our GRAMA, we consider multiple GNN benchmarks, including *MalNet-Tiny* (Freitas et al., 2021), the five heterophilic tasks introduced in (Platonov et al., 2023), *OGBG-MOLHIV* (Hu et al., 2020a), *Cora*, *CiteSeer*, *PubMed* (Yang et al., 2016), *ZINC-12k*, *MNIST*, *CIFAR10*, *PATTERN*, and *CLUSTER* (Dwivedi et al., 2023). Specifically, MalNet-Tiny consists of relatively large graphs (with thousands of nodes) representing function call graphs from malicious and benign software, where nodes represent functions and edges represent calls between them. Considering the scale of the graphs and the fact that malware can often exhibit non-local behavior, we believe this task can further reinforce the idea that GRAMA can preserve and leverage long-range interactions between nodes.

In the heterophilic node classification setting, we consider *Roman-empire, Amazon-ratings, Minesweeper, Tolokers, and Questions* tasks, to show the efficacy of our method in capturing more complex node relationships beyond simple homophily settings.

ZINC-12k and OGBG-MOLHIV are datasets where graphs represent molecules (i.e., nodes are atoms, and edges are chemical bonds) and the objective is to predict molecular properties; while Cora, CiteSeer, and PubMed are citation networks where each node represents a paper and each edge indicates that one paper cites another one, whose objective is to predict the

class associated to each node. MNIST and CIFAR10 are superpixel-based image datasets where each image is represented as a graph, and the task is to predict the digit or object class of the represented in the image; while PATTERN and CLUSTER are datasets containing stochastic block model graphs and the goal is to identify is a node belong to a pattern and assign community labels to nodes, respectively.

We consider the experimental setting from (Freitas et al., 2021) for MalNet-Tiny, and that from (Platonov et al., 2023) for heterophilic tasks. On the ZINC-12k, MNIST, CIFAR10, PATTERN, and CLUSTER, we followed the official splits and experimental protocols from (Dwivedi et al., 2023). For OGBG-MOLHIV we followed the official protocol from (Hu et al., 2020a). On Cora, CiteSeer, and PubMed, we used the splits and experimental protocols from (Pei et al., 2020). Additional details on the setup and tasks are in Appendix D.5.

**Results.** Table 5 reports the mean test set accuracy on MalNet-Tiny, Table 6 reports the test score for the heterophilic tasks, and Table 7 the test score of the other benchmarks.

Table 5: Mean test accuracy and std averaged over 4 random weight initializations on MalNet-Tiny. The higher, the better. First, second, and **third** best results. Baselines from (Wang et al., 2024a; Behrouz & Hashemi, 2024) include Laplacian positional encodings.

| Model | MalNet-Tiny Acc $\uparrow$ |
|---|---|
| **MPNNs** | |
| GCN | 81.00 |
| GIN | $88.98_{\pm 0.55}$ |
| GatedGCN | $92.23_{\pm 0.65}$ |
| ARMA | $91.80_{\pm 0.72}$ |
| **Graph Transformers** | |
| GPS+Transformer | OOM |
| GPS+Performer | $92.64_{\pm 0.78}$ |
| GPS+BigBird | $92.34_{\pm 0.34}$ |
| Exphormer | $\mathbf{94.22}_{\pm 0.24}$ |
| **Graph SSMs** | |
| Graph-Mamba | $93.40_{\pm 0.27}$ |
| GMN | **94.15** |
| **Ours** | |
| GRAMA$_{\text{GCN}}$ | $93.43_{\pm 0.29}$ |
| GRAMA$_{\text{GATEDGCN}}$ | $93.66_{\pm 0.40}$ |
| GRAMA$_{\text{GPS}}$ | $\mathbf{94.37}_{\pm 0.36}$ |

For the heterophilic tasks we included baseline results from (Finkelshtein et al., 2024; Behrouz & Hashemi, 2024; Platonov et al., 2023; Müller et al., 2024; Luan et al., 2024; Deng et al., 2024). Among all models and tasks, GRAMA achieves competitive overall performance that often outperforms state-of-the-art methods, demonstrating that our model not only excels at handling larger graphs than those considered in previous experiments but also under complex heterophilic scenarios. The results underscore the ability of GRAMA to capture the dependencies characterizing malware detection tasks where non-local behaviors are often prevalent. Overall, these findings confirm that GRAMA is a competitive and effective solution, even when compared to state-of-the-art models like Graph Transformers and recent Graph SSM models.

### E.2. Extended LRGB Comparisons

In Table 8, we report the complete results for the LRGB tasks, including more multi-hop DGNs and ablating on the scores obtained with the original setting from (Dwivedi et al., 2022b) and the one proposed in (Tönshoff et al., 2023), which leverage added residual connections and 3-layers MLP as a decoder to map the GNN output into the final prediction. In the table, we color the top three methods. Different from the main body of the paper, here, we color the best methods, including sub-variants of methods, for an additional perspective on the results.

### E.3. Complexity and Runtimes

We provide runtimes for GRAMA alongside other methods, such as Graph GPS and GCN, in Tables 9 and 10. In all cases, we use a model with 256 hidden dimensions and a varying depth (changing the sequence length $L$ from 2 to 16 in our GRAMA with $S = 2$ GRAMA blocks, recall that GRAMA depth is $SL$, and the number of layers is the backbone for other methods) and report the training and inference times, as well as the performance on the Roman-Empire dataset, for

Table 6: Mean test set score and std averaged over 4 random weight initializations on heterophilic datasets. The higher, the better. **First**, second, and **third** best results for each task are color-coded. Baseline results are reported from (Finkelshtein et al., 2024; Behrouz & Hashemi, 2024; Platonov et al., 2023; Müller et al., 2024; Luan et al., 2024; Deng et al., 2024).

| Model | Roman-empire Acc ↑ | Amazon-ratings Acc ↑ | Minesweeper AUC ↑ | Tolokers AUC ↑ | Questions AUC ↑ |
|---|---|---|---|---|---|
| **(Luan et al., 2024)** | | | | | |
| MLP-2 | $66.04_{\pm0.71}$ | $49.55_{\pm0.81}$ | $50.92_{\pm1.25}$ | $74.58_{\pm0.75}$ | $69.97_{\pm1.16}$ |
| SGC-1 | $44.60_{\pm0.52}$ | $40.69_{\pm0.42}$ | $82.04_{\pm0.77}$ | $73.80_{\pm1.35}$ | $71.06_{\pm0.92}$ |
| MLP-1 | $64.12_{\pm0.61}$ | $38.60_{\pm0.41}$ | $50.59_{\pm0.83}$ | $71.89_{\pm0.82}$ | $70.33_{\pm0.96}$ |
| **Graph-agnostic** | | | | | |
| ResNet | $65.88_{\pm0.38}$ | $45.90_{\pm0.52}$ | $50.89_{\pm1.39}$ | $72.95_{\pm1.06}$ | $70.34_{\pm0.76}$ |
| ResNet+SGC | $73.90_{\pm0.51}$ | $50.66_{\pm0.48}$ | $70.88_{\pm0.90}$ | $80.70_{\pm0.97}$ | $75.81_{\pm0.96}$ |
| ResNet+adj | $52.25_{\pm0.40}$ | $51.83_{\pm0.57}$ | $50.42_{\pm0.83}$ | $78.78_{\pm1.11}$ | $75.77_{\pm1.24}$ |
| **MPNNs** | | | | | |
| ARMA | $87.11_{\pm0.38}$ | $49.94_{\pm0.30}$ | $91.64_{\pm1.21}$ | $82.29_{\pm0.97}$ | $77.75_{\pm0.85}$ |
| GAT | $80.87_{\pm0.30}$ | $49.09_{\pm0.63}$ | $92.01_{\pm0.68}$ | $83.70_{\pm0.47}$ | $77.43_{\pm1.20}$ |
| GAT-sep | $88.75_{\pm0.41}$ | $52.70_{\pm0.62}$ | $93.91_{\pm0.35}$ | $83.78_{\pm0.43}$ | $76.79_{\pm0.71}$ |
| GAT (LapPE) | $84.80_{\pm0.46}$ | $44.90_{\pm0.73}$ | $93.50_{\pm0.54}$ | $84.99_{\pm0.54}$ | $76.55_{\pm0.84}$ |
| GAT (RWSE) | $86.62_{\pm0.53}$ | $48.58_{\pm0.41}$ | $92.53_{\pm0.65}$ | $85.02_{\pm0.67}$ | $77.83_{\pm1.22}$ |
| GAT (DEG) | $85.51_{\pm0.56}$ | $51.65_{\pm0.60}$ | $93.04_{\pm0.62}$ | $84.22_{\pm0.81}$ | $77.10_{\pm1.23}$ |
| Gated-GCN | $74.46_{\pm0.54}$ | $43.00_{\pm0.32}$ | $87.54_{\pm1.22}$ | $77.31_{\pm1.14}$ | $76.61_{\pm1.13}$ |
| GCN | $73.69_{\pm0.74}$ | $48.70_{\pm0.63}$ | $89.75_{\pm0.52}$ | $83.64_{\pm0.67}$ | $76.09_{\pm1.27}$ |
| GCN (LapPE) | $83.37_{\pm0.55}$ | $44.35_{\pm0.36}$ | $94.26_{\pm0.49}$ | $84.95_{\pm0.78}$ | $77.79_{\pm1.34}$ |
| GCN (RWSE) | $84.84_{\pm0.55}$ | $46.40_{\pm0.55}$ | $93.84_{\pm0.48}$ | $85.11_{\pm0.77}$ | $77.81_{\pm1.40}$ |
| GCN (DEG) | $84.21_{\pm0.47}$ | $50.01_{\pm0.69}$ | $94.14_{\pm0.50}$ | $82.51_{\pm0.83}$ | $76.96_{\pm1.21}$ |
| CO-GNN$(\Sigma, \Sigma)$ | $91.57_{\pm0.32}$ | $51.28_{\pm0.56}$ | $95.09_{\pm1.18}$ | $83.36_{\pm0.89}$ | $80.02_{\pm0.86}$ |
| CO-GNN$(\mu, \mu)$ | $91.37_{\pm0.35}$ | $54.17_{\pm0.37}$ | $97.31_{\pm0.41}$ | $84.45_{\pm1.17}$ | $76.54_{\pm0.95}$ |
| SAGE | $85.74_{\pm0.67}$ | $53.63_{\pm0.39}$ | $93.51_{\pm0.57}$ | $82.43_{\pm0.44}$ | $76.44_{\pm0.62}$ |
| **Graph Transformers** | | | | | |
| Exphormer | $89.03_{\pm0.37}$ | $53.51_{\pm0.46}$ | $90.74_{\pm0.53}$ | $83.77_{\pm0.78}$ | $73.94_{\pm1.06}$ |
| NAGphormer | $74.34_{\pm0.77}$ | $51.26_{\pm0.72}$ | $84.19_{\pm0.66}$ | $78.32_{\pm0.95}$ | $68.17_{\pm1.53}$ |
| GOAT | $71.59_{\pm1.25}$ | $44.61_{\pm0.50}$ | $81.09_{\pm1.02}$ | $83.11_{\pm1.04}$ | $75.76_{\pm1.66}$ |
| GPS | $82.00_{\pm0.61}$ | $53.10_{\pm0.42}$ | $90.63_{\pm0.67}$ | $83.71_{\pm0.48}$ | $71.73_{\pm1.47}$ |
| GPS$_{GCN+Performer}$ (LapPE) | $83.96_{\pm0.53}$ | $48.20_{\pm0.67}$ | $93.85_{\pm0.41}$ | $84.72_{\pm0.77}$ | $77.85_{\pm1.25}$ |
| GPS$_{GCN+Performer}$ (RWSE) | $84.72_{\pm0.65}$ | $48.08_{\pm0.85}$ | $92.88_{\pm0.50}$ | $84.81_{\pm0.86}$ | $76.45_{\pm1.51}$ |
| GPS$_{GCN+Performer}$ (DEG) | $83.38_{\pm0.68}$ | $48.93_{\pm0.47}$ | $93.60_{\pm0.47}$ | $80.49_{\pm0.97}$ | $74.24_{\pm1.18}$ |
| GPS$_{GAT+Performer}$ (LapPE) | $85.93_{\pm0.52}$ | $48.86_{\pm0.38}$ | $92.62_{\pm0.79}$ | $84.62_{\pm0.54}$ | $76.71_{\pm0.98}$ |
| GPS$_{GAT+Performer}$ (RWSE) | $87.04_{\pm0.58}$ | $49.92_{\pm0.66}$ | $91.08_{\pm0.58}$ | $84.38_{\pm0.91}$ | $77.14_{\pm1.49}$ |
| GPS$_{GAT+Performer}$ (DEG) | $85.54_{\pm0.58}$ | $51.03_{\pm0.60}$ | $91.52_{\pm0.46}$ | $82.45_{\pm0.89}$ | $76.51_{\pm1.19}$ |
| GPS$_{GCN+Transformer}$ (LapPE) | OOM | OOM | $91.82_{\pm0.41}$ | $83.51_{\pm0.93}$ | OOM |
| GPS$_{GCN+Transformer}$ (RWSE) | OOM | OOM | $91.17_{\pm0.51}$ | $83.53_{\pm1.06}$ | OOM |
| GPS$_{GCN+Transformer}$ (DEG) | OOM | OOM | $91.76_{\pm0.61}$ | $80.82_{\pm0.95}$ | OOM |
| GPS$_{GAT+Transformer}$ (LapPE) | OOM | OOM | $92.29_{\pm0.61}$ | $84.70_{\pm0.56}$ | OOM |
| GPS$_{GAT+Transformer}$ (RWSE) | OOM | OOM | $90.82_{\pm0.56}$ | $84.01_{\pm0.96}$ | OOM |
| GPS$_{GAT+Transformer}$ (DEG) | OOM | OOM | $91.58_{\pm0.56}$ | $81.89_{\pm0.85}$ | OOM |
| GT | $86.51_{\pm0.73}$ | $51.17_{\pm0.66}$ | $91.85_{\pm0.76}$ | $83.23_{\pm0.64}$ | $77.95_{\pm0.68}$ |
| GT-sep | $87.32_{\pm0.39}$ | $52.18_{\pm0.80}$ | $92.29_{\pm0.47}$ | $82.52_{\pm0.92}$ | $78.05_{\pm0.93}$ |
| Polynormer | $92.55_{\pm0.30}$ | $54.81_{\pm0.49}$ | $97.46_{\pm0.36}$ | $85.91_{\pm0.74}$ | $78.92_{\pm0.89}$ |
| **Heterophily-Designated GNNs** | | | | | |
| CPGNN | $63.96_{\pm0.62}$ | $39.79_{\pm0.77}$ | $52.03_{\pm5.46}$ | $73.36_{\pm1.01}$ | $65.96_{\pm1.95}$ |
| FAGCN | $65.22_{\pm0.56}$ | $44.12_{\pm0.30}$ | $88.17_{\pm0.73}$ | $77.75_{\pm1.05}$ | $77.24_{\pm1.26}$ |
| FSGNN | $79.92_{\pm0.56}$ | $52.74_{\pm0.83}$ | $90.08_{\pm0.70}$ | $82.76_{\pm0.61}$ | $78.86_{\pm0.92}$ |
| GBK-GNN | $74.57_{\pm0.47}$ | $45.98_{\pm0.71}$ | $90.85_{\pm0.58}$ | $81.01_{\pm0.67}$ | $74.47_{\pm0.86}$ |
| GloGNN | $59.63_{\pm0.69}$ | $36.89_{\pm0.14}$ | $51.08_{\pm1.23}$ | $73.39_{\pm1.17}$ | $65.74_{\pm1.19}$ |
| GPR-GNN | $64.85_{\pm0.27}$ | $44.88_{\pm0.34}$ | $86.24_{\pm0.61}$ | $72.94_{\pm0.97}$ | $55.48_{\pm0.91}$ |
| H2GCN | $60.11_{\pm0.52}$ | $36.47_{\pm0.23}$ | $89.71_{\pm0.31}$ | $73.35_{\pm1.01}$ | $63.59_{\pm1.46}$ |
| JacobiConv | $71.14_{\pm0.42}$ | $43.55_{\pm0.48}$ | $89.66_{\pm0.40}$ | $68.66_{\pm0.65}$ | $73.88_{\pm1.16}$ |
| **Graph SSMs** | | | | | |
| GMN | $87.69_{\pm0.50}$ | $54.07_{\pm0.31}$ | $91.01_{\pm0.23}$ | $84.52_{\pm0.21}$ | – |
| GPS + Mamba | $83.10_{\pm0.28}$ | $45.13_{\pm0.97}$ | $89.93_{\pm0.54}$ | $83.70_{\pm1.05}$ | – |
| **Ours** | | | | | |
| GRAMA$_{GCN}$ | $88.61_{\pm0.43}$ | $53.48_{\pm0.62}$ | $95.27_{\pm0.71}$ | $86.23_{\pm1.10}$ | $79.23_{\pm1.16}$ |
| GRAMA$_{GATEDGCN}$ | $91.82_{\pm0.39}$ | $53.71_{\pm0.57}$ | $98.19_{\pm0.58}$ | $85.42_{\pm0.95}$ | $80.47_{\pm1.09}$ |
| GRAMA$_{GPS}$ | $91.73_{\pm0.59}$ | $53.36_{\pm0.38}$ | $98.33_{\pm0.55}$ | $85.71_{\pm0.98}$ | $79.11_{\pm1.19}$ |

reference. As can be seen from the results in the Table, our GRAMA is positioned as a middle ground solution in terms of *computational* efficiency, between linear complexity MPNNs like GCN and quadratic complexity methods like GPS. Notably, our GRAMA achieves better performance than GCN and GPS, and maintains its performance as depth increases, different than GCN. Still, in some cases, lower computational cost might be a strong requirement, for example, on edge

Table 7: Mean test set score and std on popular GNN Benchmarks. First, second, and **third** best results for each task are color-coded.

| Model | ZINC-12k | OGBG-MOLHIV | Cora | CiteSeer | PubMed | MNIST | CIFAR10 | PATTERN | CLUSTER |
|---|---|---|---|---|---|---|---|---|---|
| | MAE ↓ | AUC ↑ | Acc ↑ | Acc ↑ | Acc ↑ | Acc ↑ | Acc ↑ | Acc ↑ | Acc ↑ |
| GCN | $0.278_{\pm 0.003}$ | $76.06_{\pm 0.97}$ | $85.77_{\pm 1.27}$ | $73.68_{\pm 1.36}$ | $88.13_{\pm 0.50}$ | $90.71_{\pm 0.22}$ | $55.71_{\pm 0.38}$ | $71.89_{\pm 0.33}$ | $68.50_{\pm 0.98}$ |
| GatedGCN | $0.254_{\pm 0.005}$ | $76.72_{\pm 0.88}$ | $86.21_{\pm 1.28}$ | $74.10_{\pm 1.22}$ | $88.09_{\pm 0.44}$ | $97.34_{\pm 0.14}$ | $67.31_{\pm 0.31}$ | $85.57_{\pm 0.09}$ | $73.84_{\pm 0.33}$ |
| GPS | $0.125_{\pm 0.009}$ | $77.39_{\pm 1.14}$ | $85.42_{\pm 1.80}$ | $73.99_{\pm 1.57}$ | $88.23_{\pm 0.61}$ | $98.05_{\pm 0.13}$ | $72.30_{\pm 0.36}$ | $86.69_{\pm 0.06}$ | $78.02_{\pm 0.18}$ |
| GPS + RWSE | $\mathbf{0.070}_{\pm 0.004}$ | $78.80_{\pm 1.01}$ | $86.67_{\pm 1.53}$ | $74.52_{\pm 1.49}$ | $88.94_{\pm 0.49}$ | - | - | - | - |
| EGT | $0.108_{\pm 0.009}$ | - | - | - | - | $98.17_{\pm 0.09}$ | $68.70_{\pm 0.41}$ | $86.82_{\pm 0.02}$ | $79.23_{\pm 0.35}$ |
| GRIT | $0.059_{\pm 0.002}$ | - | - | - | - | $98.11_{\pm 0.11}$ | $76.47_{\pm 0.88}$ | $87.20_{\pm 0.08}$ | $80.03_{\pm 0.28}$ |
| **Ours** | | | | | | | | | |
| GRAMA$_{\text{GCN}}$ | $0.142_{\pm 0.010}$ | $77.47_{\pm 1.05}$ | $\mathbf{88.02}_{\pm 1.01}$ | $77.09_{\pm 1.53}$ | $90.20_{\pm 0.47}$ | $97.87_{\pm 0.19}$ | $70.28_{\pm 0.42}$ | $82.66_{\pm 0.18}$ | $74.29_{\pm 0.60}$ |
| GRAMA$_{\text{GatedGCN}}$ | $0.140_{\pm 0.008}$ | $77.60_{\pm 0.98}$ | $88.13_{\pm 0.99}$ | $\mathbf{77.63}_{\pm 1.38}$ | $90.07_{\pm 0.45}$ | $98.12_{\pm 0.10}$ | $74.61_{\pm 0.45}$ | $86.72_{\pm 0.10}$ | $76.88_{\pm 0.32}$ |
| GRAMA$_{\text{GPS}}$ | $0.100_{\pm 0.006}$ | $78.19_{\pm 1.10}$ | $87.95_{\pm 1.72}$ | $77.13_{\pm 1.51}$ | $89.76_{\pm 0.64}$ | $98.29_{\pm 0.14}$ | $75.92_{\pm 0.41}$ | $87.41_{\pm 0.07}$ | $79.66_{\pm 0.19}$ |
| GRAMA$_{\text{GPS+RWSE}}$ | $0.061_{\pm 0.003}$ | $79.21_{\pm 0.94}$ | $88.37_{\pm 1.64}$ | $77.68_{\pm 1.55}$ | $90.31_{\pm 0.58}$ | - | - | - | - |

devices. To this end, we can use the *naive* learning approach of ARMA coefficients, as discussed in Section 3.2, which still utilizes our GRAMA but avoids the use of an attention mechanism for a selective ARMA coefficient learning. In this case, as we show in Table 9, it is still possible to obtain significant improvement compared with the baseline performance with lower computational time, although with less flexibility that is offered by our selective ARMA coefficient learning. In addition, for a broader comparison, we consider the best-performing variant of GPS (GPS$_{\text{GAT+Performer}}$ (RWSE)), showing that also in this case, our GRAMA offers better performance. Furthermore, the results in Table 9 offer comparisons of GRAMA, GCN, and GPS under equivalent runtime budgets, showing that GRAMA matches or exceeds the accuracy of GPS while being more efficient. Additionally, the non-selective GRAMA variant maintains high performance with further reductions in computational cost, showcasing its applicability of GRAMA constrained computational scenarios. Finally, in Table 11 we compare our GRAMA with recent state-of-the-art methods in terms of both time and downstream performance. Our GRAMA requires similar time to other methods like GPS+Mamba and GMN, which are also selective models, while requiring significantly less time than GPS. All runtimes are measured on an NVIDIA A6000 GPU with 48GB of memory.

### E.4. Ablation Studies

To provide a comprehensive understanding of the different components and hyperparameters of our GRAMA, we now present several ablations studies.

**Selective vs. Naive ARMA Coefficients.** In Section 3.2, we describe a novel, selective way to predict the ARMA coefficients that govern our GRAMA model, as described in Section 3. We now empirically check whether the added flexibility and adaptivity help in practice. To do that, we present the results with 'naively' learned ARMA coefficients, a variant denoted by GRAMA (Naive), and also report the results with our GRAMA model (these are the same results presented in the rest of our experiments). The results are presented in Table 12. The results show that (i) our GRAMA, as an architecture, regardless of the use of selective ARMA coefficients or not, significantly improves the baseline (GCN), and (ii) learning selective ARMA coefficients offers further performance gains compared with naive coefficients.

**Performance vs. Model Depth.** We evaluate the performance of our GRAMA on varying depths. The depth is influenced by the number of recurrences $R$ and $S$ GRAMA blocks. As discussed in Section 3, to reduce the number of hyperparameters and obtain a large prediction window with respect to the input sequence, we choose $R = L$. That is, the number of recurrent steps is $L$. Thus, the effective depth of the model is the multiplication $S \cdot L$. Therefore, we test the performance of GRAMA with varying depths, up to a depth of 128 layers, and maintain a constant width of 256. The results reported in Table 13 demonstrate the ability of GRAMA to maintain and improve its performance with more layers.

**Hyperparameter Influence.** In Table 13 we showed the performance of GRAMA under varying depths. However, note that this study also shows the influence of the number of recurrences $R$ (which is also the length of the sequence $L$) and the number of GRAMA blocks $S$. The results show that both are beneficial as increased in terms of added performance, and that enlarging to a value larger than 4 maintains consistent results.

Furthermore, in Table 14, we show the performance of GRAMA with a varying with (i.e., number of hidden channels), when choosing the other hyperparameters to be fixed, and in particular $S = L = 4$. From this experiment, we can see that while some configurations offer better performance than others, overall, our GRAMA consistently improves the baseline

Table 8: Results for Peptides-func and Peptides-struct averaged over 3 training seeds. Baseline results are taken from (Dwivedi et al., 2022b) and (Gutteridge et al., 2023). Re-evaluated methods employ the 3-layer MLP readout proposed in (Tönshoff et al., 2023). Note that all MPNN-based methods include structural and positional encoding. The **first**, **second**, and **third** best scores are colored. Baseline results are reported from (Gutteridge et al., 2023; Tönshoff et al., 2023; Ma et al., 2023; Ding et al., 2024; Huang et al., 2024b; Wang et al., 2024a; Behrouz & Hashemi, 2024; Gravina et al., 2025; Heilig et al., 2025). ‡ means 3-layer MLP readout and residual connections are employed.

| Model | Peptides-func AP ↑ | Peptides-struct MAE ↓ |
|---|---|---|
| **MPNNs** | | |
| GCN | $59.30_{\pm 0.23}$ | $0.3496_{\pm 0.0013}$ |
| GINE | $54.98_{\pm 0.79}$ | $0.3547_{\pm 0.0045}$ |
| GCNII | $55.43_{\pm 0.78}$ | $0.3471_{\pm 0.0010}$ |
| GatedGCN | $58.64_{\pm 0.77}$ | $0.3420_{\pm 0.0013}$ |
| ARMA | $64.08_{\pm 0.62}$ | $0.2709_{\pm 0.0016}$ |
| **Multi-hop GNNs** | | |
| DIGL+MPNN | $64.69_{\pm 0.19}$ | $0.3173_{\pm 0.0007}$ |
| DIGL+MPNN+LapPE | $68.30_{\pm 0.26}$ | $0.2616_{\pm 0.0018}$ |
| MixHop-GCN | $65.92_{\pm 0.36}$ | $0.2921_{\pm 0.0023}$ |
| MixHop-GCN+LapPE | $68.43_{\pm 0.49}$ | $0.2614_{\pm 0.0023}$ |
| DRew-GCN | $69.96_{\pm 0.76}$ | $0.2781_{\pm 0.0028}$ |
| DRew-GCN+LapPE | $\mathbf{71.50}_{\pm 0.44}$ | $0.2536_{\pm 0.0015}$ |
| DRew-GIN | $69.40_{\pm 0.74}$ | $0.2799_{\pm 0.0016}$ |
| DRew-GIN+LapPE | $71.26_{\pm 0.45}$ | $0.2606_{\pm 0.0014}$ |
| DRew-GatedGCN | $67.33_{\pm 0.94}$ | $0.2699_{\pm 0.0018}$ |
| DRew-GatedGCN+LapPE | $69.77_{\pm 0.26}$ | $0.2539_{\pm 0.0007}$ |
| GRED | $70.85_{\pm 0.27}$ | $0.2503_{\pm 0.0019}$ |
| **Transformers** | | |
| Transformer+LapPE | $63.26_{\pm 1.26}$ | $0.2529_{\pm 0.0016}$ |
| SAN+LapPE | $63.84_{\pm 1.21}$ | $0.2683_{\pm 0.0043}$ |
| GraphGPS+LapPE | $65.35_{\pm 0.41}$ | $0.2500_{\pm 0.0005}$ |
| GRIT | $69.88_{\pm 0.82}$ | $0.2460_{\pm 0.0012}$ |
| **Modified and Re-evaluated**‡ | | |
| GCN | $68.60_{\pm 0.50}$ | $0.2460_{\pm 0.0007}$ |
| GINE | $66.21_{\pm 0.67}$ | $0.2473_{\pm 0.0017}$ |
| GatedGCN | $67.65_{\pm 0.47}$ | $0.2477_{\pm 0.0009}$ |
| DRew-GCN+LapPE | $69.45_{\pm 0.21}$ | $0.2517_{\pm 0.0011}$ |
| GraphGPS+LapPE | $65.34_{\pm 0.91}$ | $0.2509_{\pm 0.0014}$ |
| PH-DGN | $70.12_{\pm 0.45}$ | $0.2465_{\pm 0.0020}$ |
| **DE-GNNs** | | |
| GRAND | $57.89_{\pm 0.62}$ | $0.3418_{\pm 0.0015}$ |
| GraphCON | $60.22_{\pm 0.68}$ | $0.2778_{\pm 0.0018}$ |
| A-DGN | $59.75_{\pm 0.44}$ | $0.2874_{\pm 0.0021}$ |
| SWAN | $67.51_{\pm 0.39}$ | $0.2485_{\pm 0.0009}$ |
| **Graph SSMs** | | |
| Graph-Mamba | $67.39_{\pm 0.87}$ | $0.2478_{\pm 0.0016}$ |
| GMN | $70.71_{\pm 0.83}$ | $0.2473_{\pm 0.0025}$ |
| GSSC | $70.81_{\pm 0.62}$ | $\mathbf{0.2459}_{\pm 0.0020}$ |
| **Ours** | | |
| GRAMA$_{\text{GCN}}$ | $\mathbf{70.93}_{\pm 0.78}$ | $0.2439_{\pm 0.0017}$ |
| GRAMA$_{\text{GATEDGCN}}$ | $70.49_{\pm 0.51}$ | $\mathbf{0.2459}_{\pm 0.0020}$ |
| GRAMA$_{\text{GPS}}$ | $69.83_{\pm 0.83}$ | $\mathbf{0.2436}_{\pm 0.0022}$ |

methods compared with other baselines reported in Table 6.

## E.5. Additional Comparison with MixHop

In this section, we report a deeper comparison with MixHop (Abu-El-Haija et al., 2019). While MixHop assigns learnable weights per layer and uses a dense layer with space complexity $Ld^2$, GRAMA is more efficient. In the non-selective (naive) variant, GRAMA learns only $2L$ parameters per block, which is highly scalable since $L \ll d$ and $L \ll n$. In the selective variant, parameter count depends only on $d$, with space complexity $d^2$; the $L$ dependence appears only in the $L^2$ complexity of the sequence input, as discussed in Section 3.3.

As shown in Tables 2 and 8 and Table 15, GRAMA outperforms MixHop. Moreover, our results in the ablation study in Table 12 show that even without the selective mechanism, GRAMA still performs better. This highlights that the gain comes not only from selectiveness but also from GRAMA's overall design: unlike MixHop's dense, non-recurrent projection,

Table 9: Training and Inference Runtime (milliseconds) and obtained node classification accuracy (%) on the Roman-Empire dataset. Note that in $\text{GRAMA}_{\text{GCN}}$ (Naive), the ARMA coefficients are learned, but not input-adaptive as in $\text{GRAMA}_{\text{GCN}}$.

| Metrics | Method | Depth | | | |
|---|---|---|---|---|---|
| | | 4 | 8 | 16 | 32 |
| Training (ms) | | 18.38 | 33.09 | 61.86 | 120.93 |
| Inference (ms) | GCN | 9.30 | 14.64 | 27.95 | 53.55 |
| Accuracy (%) | | 73.60 | 61.52 | 56.86 | 52.42 |
| Training (ms) | | 1139.05 | 2286.96 | 4545.46 | OOM |
| Inference (ms) | GPS | 119.10 | 208.26 | 427.89 | OOM |
| Accuracy (%) | | 81.97 | 81.53 | 81.88 | OOM |
| Training (ms) | | 1179.08 | 2304.77 | 4590.26 | OOM |
| Inference (ms) | $\text{GPS}_{\text{GAT+Performer}}$ (RWSE) | 120.11 | 209.98 | 429.03 | OOM |
| Accuracy (%) | | 84.89 | 87.01 | 86.94 | OOM |
| Training (ms) | | 41.16 | 98.83 | 249.68 | 747.26 |
| Inference (ms) | $\text{GRAMA}_{\text{GCN}}$ (Naive) | 13.03 | 26.83 | 63.61 | 164.87 |
| Accuracy (%) | | 83.23 | 84.72 | 85.13 | 85.04 |
| Training (ms) | | 75.75 | 141.79 | 463.76 | 1378.91 |
| Inference (ms) | $\text{GRAMA}_{\text{GCN}}$ | 40.33 | 70.91 | 240.78 | 702.17 |
| Accuracy (%) | | 86.33 | 88.14 | 88.24 | 88.22 |

Table 10: Training Runtime (milliseconds) on the Roman-Empire dataset of GRAMA and its backbone GNNs. Note that in $\text{GRAMA}_{\text{GCN}}$ (Naive), the ARMA coefficients are learned, but not input-adaptive as in $\text{GRAMA}_{\text{GCN}}$.

| Method | Depth | | | |
|---|---|---|---|---|
| | 4 | 8 | 16 | 32 |
| GCN | 18.38 | 33.09 | 61.86 | 120.93 |
| GatedGCN | 27.57 | 47.98 | 85.36 | 171.27 |
| GPS | 1139.05 | 2286.96 | 4545.46 | OOM |
| $\text{GRAMA}_{\text{GCN}}$ (Naive) | 41.16 | 98.83 | 249.68 | 747.26 |
| $\text{GRAMA}_{\text{GCN}}$ | 75.75 | 141.79 | 463.76 | 1378.91 |
| $\text{GRAMA}_{\text{GATEDGCN}}$ | 51.49 | 117.01 | 270.64 | 792.32 |
| $\text{GRAMA}_{\text{GPS}}$ | 1162.13 | 2346.94 | 4642.19 | OOM |

GRAMA uses a recurrent, non-dense aggregation inspired by dynamical systems and modern RNNs.

### E.6. Additional Spatio-Temporal Benchmarks

Given the inspiration of GRAMA from sequential models, by transforming a graph into sequences of the graph, it is interesting to understand if it can be utilized for spatio-temporal datasets. In this section we preliminary results with datasets from Rozemberczki et al. (2021). Specifically, we employ Chickenpox Hungary, PedalMe London, and Wikipedia Math, where the goal is to predict future values given past values. In Table 16, we compare the MSE score of our method with two state-of-the-art approaches in the spatio-temporal domain, i.e., A3T-GCN (Bai et al., 2021) and T-GCN (Zhao et al., 2020). Our results show that our GRAMA is a promising approach for processing spatio-temporal data as well.

It is important to note that addressing spatio-temporal datasets is not the main goal of this paper. Rather, our GRAMA addresses fundamental issues in existing methods that utilize graph SSMs and the oversquashing issue in static graphs, and studying and extending our GRAMA to spatio-temoporal datasets is an interesting future work direction.

## F. Summary of the results

In this section, we provide a summary of the results achieved by GRAMA. Specifically, we report Table 17 the performance of our GRAMA with respect to the baseline backbone GNNs (i.e., GCN, GatedGCN, and GPS)) and in Table 18 the comparison with the best baseline out of all methods in tables. As evidenced from both Table 17 and Table 18, our GRAMA

Table 11: Training runtime per epoch (milliseconds) and obtained node classification accuracy (%) on the Roman-Empire dataset. Note that in GRAMA$_{\text{GCN}}$ (Naive), the ARMA coefficients are learned, but not input-adaptive as in GRAMA$_{\text{GCN}}$.

| Model | Training runtime per epoch (ms) | Accuracy (%) |
|---|---|---|
| GatedGCN | 18.38 | $73.69_{\pm 0.74}$ |
| GPS | 1139.05 | $82.00_{\pm 0.61}$ |
| GPS + Mamba | 320.39 | $83.10_{\pm 0.28}$ |
| GMN | 387.04 | $87.69_{\pm 0.50}$ |
| GRAMA$_{\text{GCN}}$ (Naive) | 249.68 | $85.13_{\pm 0.36}$ |
| GRAMA$_{\text{GCN}}$ | 362.41 | $88.61_{\pm 0.43}$ |

Table 12: The significance of learning selective ARMA coefficients. Our GRAMA architectures improve baseline performance, and its selective mechanism further improves performance. Note that in GRAMA$_{\text{GCN}}$ (Naive), the ARMA coefficients are learned, but not input-adaptive as in GRAMA$_{\text{GCN}}$.

| Model | Roman-empire Acc ↑ | Peptides-func AP ↑ |
|---|---|---|
| GCN | $73.69_{\pm 0.74}$ | $59.30_{\pm 0.23}$ |
| GRAMA$_{\text{GCN}}$ (Naive) | $85.13_{\pm 0.58}$ | $68.98_{\pm 0.52}$ |
| GRAMA$_{\text{GCN}}$ | $88.61_{\pm 0.43}$ | $70.93_{\pm 0.78}$ |

offers significant and consistent improvements over the baseline backbone GNNs as well as better performance than current state-of-the-art performing methods. Therefore, we believe that our proposed method provides a substantial improvement not only in terms of designing a principled and mathematically sound model, which is equivalent to a Graph SSM model that preserves permutation-equivariance and allows long-range propagation, but also in terms of downstream performance on real-world applications and benchmarks.

Table 13: The obtained node classification accuracy (%) with GRAMA$_{GCN}$ on the Roman-Empire with a varying sequence length size $L$ and blocks $S$. Our GRAMA improves with more layers, and maintains its performance with deep models.

| Sequence Length $L \downarrow$ / Blocks $S \rightarrow$ | 2 | 4 | 8 | 16 |
|---|---|---|---|---|
| 2 | 86.33 | 88.14 | 88.24 | 88.22 |
| 4 | 87.30 | 88.06 | 88.61 | 88.57 |
| 8 | 88.41 | 88.15 | 88.54 | 88.46 |

Table 14: Node classification accuracy (%) on Roman-Empire with varying width of GRAMA.

| Model $\downarrow$ / Width $\rightarrow$ | 64 | 128 | 256 |
|---|---|---|---|
| GRAMA$_{GCN}$ | 87.79 | 88.45 | 88.61 |
| GRAMA$_{GATEDGCN}$ | 91.79 | 91.66 | 91.68 |
| GRAMA$_{GPS}$ | 91.28 | 91.70 | 91.19 |

Table 15: Mean test set score and std on popular benchmarks comparing MixHop and GRAMA.

| Model | Peptides-func AP ↑ | Peptides-struct MAE ↓ | Roman-Empire Acc ↑ | Amazon-Ratings Acc ↑ | OGBN-Arxiv Acc ↑ |
|---|---|---|---|---|---|
| MixHop-GCN | $68.43_{\pm 0.49}$ | $0.2614_{\pm 0.0023}$ | $79.16_{\pm 0.70}$ | $47.95_{\pm 0.65}$ | $71.29_{\pm 0.29}$ |
| GRAMA$_{GCN}$ | $\mathbf{70.93}_{\pm 0.78}$ | $\mathbf{0.2439}_{\pm 0.0017}$ | $\mathbf{88.61}_{\pm 0.43}$ | $\mathbf{53.48}_{\pm 0.62}$ | $\mathbf{73.86}_{\pm 0.21}$ |

Table 16: Mean test set MSE and std on spatio-temporal datasets. The **best** result for each task is color-coded.

| Model | Chickenpox Hungary | PedalMe London | Wikipedia Math |
|---|---|---|---|
| **Baselines** | | | |
| A3T-GCN | $1.114_{\pm 0.008}$ | $1.469_{\pm 0.027}$ | $0.781_{\pm 0.011}$ |
| T-GCN | $1.117_{\pm 0.011}$ | $1.479_{\pm 0.012}$ | $0.764_{\pm 0.011}$ |
| **Our** | | | |
| GRAMA$_{GCN}$ | $\mathbf{0.790}_{\pm 0.031}$ | $\mathbf{1.089}_{\pm 0.049}$ | $\mathbf{0.608}_{\pm 0.019}$ |

Table 17: Summary of the performance of our GRAMA with respect to backbone GNNs. The **best** result for each task is color-coded.

| Task $\downarrow$ / Model $\rightarrow$ | GCN | GatedGCN | GPS | Ours GRAMA$_{GCN}$ | GRAMA$_{GATEDGCN}$ | GRAMA$_{GPS}$ |
|---|---|---|---|---|---|---|
| Diameter ($log_{10}(MSE)$ ↓) | $0.7424_{\pm 0.0466}$ | $0.1348_{\pm 0.0397}$ | $-0.5121_{\pm 0.0426}$ | $0.2577_{\pm 0.0368}$ | $-0.5485_{\pm 0.1489}$ | $\mathbf{-0.8663}_{\pm 0.0514}$ |
| SSSP ($log_{10}(MSE)$ ↓) | $0.9499_{\pm 9.18 \cdot 10^{-5}}$ | $-3.261_{\pm 0.0514}$ | $-3.599_{\pm 0.1949}$ | $0.0095_{\pm 0.0877}$ | $\mathbf{-4.1289}_{\pm 0.0988}$ | $-3.9349_{\pm 0.0699}$ |
| Ecc. ($log_{10}(MSE)$ ↓) | $0.8468_{\pm 0.0028}$ | $0.6995_{\pm 0.0302}$ | $0.6077_{\pm 0.0282}$ | $0.6193_{\pm 0.0441}$ | $0.5523_{\pm 0.0511}$ | $\mathbf{-1.3012}_{\pm 0.1258}$ |
| Pept.-func (AP ↑) | $59.30_{\pm 0.23}$ | $58.64_{\pm 0.77}$ | $65.35_{\pm 0.41}$ | $\mathbf{70.93}_{\pm 0.78}$ | $70.49_{\pm 0.51}$ | $69.83_{\pm 0.83}$ |
| Pept.-struct (MAE ↓) | $0.3496_{\pm 0.0013}$ | $0.3420_{\pm 0.0013}$ | $0.2500_{\pm 0.0005}$ | $0.2439_{\pm 0.0017}$ | $0.2459_{\pm 0.0020}$ | $\mathbf{0.2436}_{\pm 0.0022}$ |
| MalNet-Tiny (Acc ↑) | $81.00$ | $92.23_{\pm 0.65}$ | $92.64_{\pm 0.78}$ | $93.43_{\pm 0.29}$ | $93.66_{\pm 0.40}$ | $\mathbf{94.37}_{\pm 0.36}$ |
| Roman-empire (Acc ↑) | $73.69_{\pm 0.74}$ | $74.46_{\pm 0.54}$ | $82.00_{\pm 0.61}$ | $88.61_{\pm 0.43}$ | $\mathbf{91.82}_{\pm 0.39}$ | $91.73_{\pm 0.59}$ |
| Amazon-ratings (Acc ↑) | $48.70_{\pm 0.63}$ | $43.00_{\pm 0.32}$ | $53.10_{\pm 0.42}$ | $53.48_{\pm 0.62}$ | $\mathbf{53.71}_{\pm 0.57}$ | $53.36_{\pm 0.38}$ |
| Minesweeper (AUC ↑) | $89.75_{\pm 0.52}$ | $87.54_{\pm 1.22}$ | $90.63_{\pm 0.67}$ | $95.27_{\pm 0.71}$ | $98.19_{\pm 0.58}$ | $\mathbf{98.33}_{\pm 0.55}$ |
| Tolokers (AUC ↑) | $83.64_{\pm 0.67}$ | $77.31_{\pm 1.14}$ | $83.71_{\pm 0.48}$ | $\mathbf{86.23}_{\pm 1.10}$ | $85.42_{\pm 0.95}$ | $85.71_{\pm 0.98}$ |
| Questions (AUC ↑) | $76.09_{\pm 1.27}$ | $76.61_{\pm 1.13}$ | $71.73_{\pm 1.47}$ | $79.23_{\pm 1.16}$ | $\mathbf{80.47}_{\pm 1.09}$ | $79.11_{\pm 1.19}$ |
| ZINC-12k (MAE ↓) | $0.278_{\pm 0.003}$ | $0.254_{\pm 0.005}$ | $0.125_{\pm 0.009}$ | $0.142_{\pm 0.010}$ | $0.140_{\pm 0.008}$ | $\mathbf{0.100}_{\pm 0.006}$ |
| OGBG-MOLHIV (AUC ↑) | $76.06_{\pm 0.97}$ | $76.72_{\pm 0.88}$ | $77.39_{\pm 1.14}$ | $77.47_{\pm 1.05}$ | $77.60_{\pm 0.98}$ | $\mathbf{78.19}_{\pm 1.10}$ |
| Cora (Acc ↑) | $85.77_{\pm 1.27}$ | $86.21_{\pm 1.28}$ | $85.42_{\pm 1.80}$ | $88.02_{\pm 1.01}$ | $\mathbf{88.13}_{\pm 0.99}$ | $87.95_{\pm 1.72}$ |
| CiteSeer (Acc ↑) | $73.68_{\pm 1.36}$ | $74.10_{\pm 1.22}$ | $73.99_{\pm 1.57}$ | $77.09_{\pm 1.53}$ | $\mathbf{77.63}_{\pm 1.38}$ | $77.13_{\pm 1.51}$ |
| PubMed (Acc ↑) | $88.13_{\pm 0.50}$ | $88.09_{\pm 0.44}$ | $88.23_{\pm 0.61}$ | $\mathbf{90.20}_{\pm 0.47}$ | $90.07_{\pm 0.45}$ | $89.76_{\pm 0.64}$ |
| MNIST (Acc ↑) | $90.71_{\pm 0.22}$ | $97.34_{\pm 0.14}$ | $98.05_{\pm 0.13}$ | $97.87_{\pm 0.19}$ | $98.12_{\pm 0.10}$ | $\mathbf{98.29}_{\pm 0.14}$ |
| CIFAR10 (Acc ↑) | $55.71_{\pm 0.38}$ | $67.31_{\pm 0.31}$ | $72.30_{\pm 0.36}$ | $70.28_{\pm 0.42}$ | $74.61_{\pm 0.45}$ | $\mathbf{75.92}_{\pm 0.41}$ |
| PATTERN (Acc ↑) | $71.89_{\pm 0.33}$ | $85.57_{\pm 0.09}$ | $86.69_{\pm 0.06}$ | $82.66_{\pm 0.18}$ | $86.72_{\pm 0.10}$ | $\mathbf{87.41}_{\pm 0.07}$ |
| CLUSTER (Acc ↑) | $68.50_{\pm 0.98}$ | $73.84_{\pm 0.33}$ | $78.02_{\pm 0.18}$ | $74.29_{\pm 0.60}$ | $76.88_{\pm 0.32}$ | $\mathbf{79.66}_{\pm 0.19}$ |

Table 18: Summary of the performance of our GRAMA (best performing model out of 3 variants) with respect to the best baseline out of all methods in Tables. The **best** results for each task is color-coded. The "Improvement" column reports the difference in performance between GRAMA and the best baseline

| Task ↓ / Model → | Best baseline | GRAMA | Improvement |
|---|---|---|---|
| Diameter $(log_{10}(MSE)\downarrow)$ | $-0.5981_{\pm 0.1145}$ | $\mathbf{-0.8663}_{\pm 0.0514}$ | -0.2682 |
| SSSP $(log_{10}(MSE)\downarrow)$ | $-3.5990_{\pm 0.1949}$ | $\mathbf{-4.1289}_{\pm 0.0988}$ | -0.5299 |
| Ecc. $(log_{10}(MSE)\downarrow)$ | $-0.0739_{\pm 0.2190}$ | $\mathbf{-1.3012}_{\pm 0.1258}$ | -1.2273 |
| Pept.-func (AP ↑) | $\mathbf{71.50}_{\pm 0.44}$ | $70.93_{\pm 0.78}$ | -0.57 |
| Pept.-struct (MAE ↓) | $0.2459_{\pm 0.0020}$ | $\mathbf{0.2436}_{\pm 0.0022}$ | 0.0023 |
| MalNet-Tiny (Acc ↑) | $94.22_{\pm 0.24}$ | $\mathbf{94.37}_{\pm 0.36}$ | 0.15 |
| Roman-empire (Acc ↑) | $\mathbf{92.55}_{\pm 0.30}$ | $\mathbf{91.82}_{\pm 0.39}$ | -0.73 |
| Amazon-ratings (Acc ↑) | $\mathbf{54.81}_{\pm 0.49}$ | $53.71_{\pm 0.57}$ | -1.08 |
| Minesweeper (AUC ↑) | $97.46_{\pm 0.36}$ | $\mathbf{98.33}_{\pm 0.55}$ | 0.87 |
| Tolokers (AUC ↑) | $85.91_{\pm 0.74}$ | $\mathbf{86.23}_{\pm 1.10}$ | 0.32 |
| Questions (AUC ↑) | $80.02_{\pm 0.86}$ | $\mathbf{80.47}_{\pm 1.09}$ | 0.45 |
| ZINC-12k (MAE ↓) | $\mathbf{0.059}_{\pm 0.002}$ | $0.061_{\pm 0.003}$ | 0.002 |
| OGBG-MOLHIV (AUC ↑) | $78.80_{\pm 1.01}$ | $\mathbf{79.21}_{\pm 0.94}$ | 0.41 |
| Cora (Acc ↑) | $86.67_{\pm 1.53}$ | $\mathbf{88.37}_{\pm 1.64}$ | 1.7 |
| CiteSeer (Acc ↑) | $74.52_{\pm 1.49}$ | $\mathbf{77.68}_{\pm 1.55}$ | 3.16 |
| PubMed (Acc ↑) | $88.94_{\pm 0.49}$ | $\mathbf{90.31}_{\pm 0.58}$ | 1.37 |
| MNIST (Acc ↑) | $98.17_{\pm 0.09}$ | $\mathbf{98.29}_{\pm 0.14}$ | 0.12 |
| CIFAR10 (Acc ↑) | $\mathbf{76.47}_{\pm 0.88}$ | $75.92_{\pm 0.41}$ | 0.55 |
| PATTERN (Acc ↑) | $87.20_{\pm 0.08}$ | $\mathbf{87.41}_{\pm 0.07}$ | 0.21 |
| CLUSTER (Acc ↑) | $\mathbf{80.03}_{\pm 0.28}$ | $79.66_{\pm 0.19}$ | 0.37 |
| Chickenpox Hungary (MSE ↓) | $1.114_{\pm 0.008}$ | $\mathbf{0.790}_{\pm 0.031}$ | -0.324 |
| PedalMe London (MSE ↓) | $1.469_{\pm 0.027}$ | $\mathbf{1.089}_{\pm 0.049}$ | -0.380 |
| Wikipedia Math (MSE ↓) | $0.764_{\pm 0.011}$ | $\mathbf{0.608}_{\pm 0.019}$ | -0.156 |

