# OpenReview forum: "Graph Adaptive Autoregressive Moving Average Models"
_ICML.cc/2025/Conference — ICML 2025 spotlightposter_

### Official Review · Reviewer_NJt1 · 2025-03-09

**Overall Recommendation:** 3

**Summary:**

In this paper, the authors propose addressing the computational complexity issues in traditional graph transformers and the over-squashing problem in GNNs by converting the input graph into a sequential representation and incorporating an autoregressive moving average model with an attention selection mechanism.

**Claims And Evidence:**

The authors argue that the biggest challenge in applying sequence models to graph-structured data is `how to transform a graph into a sequential representation`. However, their discussion is limited to the scope of SSM. In fact, research on converting graph data into regular or Euclidean-structured data has long been explored in the graph community, as seen in studies R1–R6.

Additionally, the authors employ `recurrent layers and dynamical system` designs in their method. Similar approaches have also been investigated in the community, such as recurrent layer methods (R5, R7, R8) and implicit model approaches (R9, R10).

The authors should consider incorporating these works into the discussion.

R1. Learning convolutional neural networks for graphs. ICML 2016

R2. Janossy Pooling: Learning Deep Permutation-Invariant Functions for Variable-Size Inputs. ICLR 2019

R3. Relational Pooling for Graph Representations. ICML 2019

R4. pathGCN: Learning General Graph Spatial Operators from Paths. ICML 2022

R5. Going Deeper into Permutation-Sensitive Graph Neural Networks. ICML 2022

R6. All in a row: Compressed Convolution Networks for Graphs. ICML 2023

R7. The Graph Neural Network Model. Trans. Neural Networks 2009

R8. Towards Dynamic Message Passing on Graphs. NeurIPS 2024

R9. Implicit Graph Neural Networks. NeurIPS 2020

R10. Optimization-Induced Graph Implicit Nonlinear Diffusion. ICML 2022

**Essential References Not Discussed:**

Please refer to the claims and evidence.

**Experimental Designs Or Analyses:**

The experiments in this paper are relatively comprehensive. In addition to the main content, the appendix provides a substantial amount of supplementary experimental results. However, there are still some issues with the experimental designs and analyses.
- One of the motivations of this paper is to address the time consumption issue caused by global attention in graph transformers. Many subsequent GT methods have already attempted to tackle this problem (R13–R15). The authors should consider thoroughly comparing GRAMA with these methods to further substantiate its advantages.
- GRAMA relies on multiple time steps and blocks to extract features from the input graph. However, the authors only report the time performance of GRAMA+GCN. According to the experimental results, GRAMA combined with GatedGCN or GPS often achieves better results. How do these two models perform in terms of time consumption?

R13. GOAT: A Global Transformer on Large-scale Graphs. ICML 2023

R14. Polynormer: PolynomialExpressive Graph Transformer in Linear Time. ICLR 2024

R15. Exphormer: Sparse Transformers for Graphs. ICML 2023

**Methods And Evaluation Criteria:**

- The proposed method can essentially be seen as MixHop (R11) with layer-selective attention. The authors should consider comparing a simple MixHop approach with GRAMA in the ablation study to demonstrate the effectiveness of the more complex strategy proposed in this paper.
- Since GRAMA needs to maintain a feature matrix of size $L\times n\times d$, its scalability may be affected. For example, MixHop is prone to OOM issues when applied to large-scale graphs. The authors should consider incorporating more large-scale graphs as benchmarks (e.g., R12).

R11. MixHop: Higher-Order Graph Convolutional Architectures via Sparsified Neighborhood Mixing. ICML 2019

R12. Open Graph Benchmark: Datasets for Machine Learning on Graphs. NeurIPS 2020

**Other Comments Or Suggestions:**

**Post Rebuttal**: My concerns have been addressed. Based on the rebuttal, I am changing my score to 3. I hope the authors will incorporate the discussions from the rebuttal phase into the revised version of the manuscript.

**Other Strengths And Weaknesses:**

The paper aims to improve the model's ability to capture long-range information with reduced computational cost. This issue has been a long-standing focus of research in the graph community.

**Questions For Authors:**

N/A

**Relation To Broader Scientific Literature:**

N/A

**Theoretical Claims:**

There are no issues in this regard for now.

---

> ### Author Rebuttal · Authors · 2025-04-01
>
> We thank the Reviewer for the thorough and constructive feedback. We have carefully addressed all concerns and believe the revisions have strengthened our paper. We hope our responses are satisfactory and that you will consider updating your score.
>
> ---
>
> **Regarding R1-R6:** The Reviewer is correct that prior works have studied converting graphs into sequences, which we now cite in the revised paper. However, as noted, our focus is on Graph State Space Models, making those works only tangentially related. Our aim is to design a graph SSM that enables long-range processing while preserving permutation equivariance. For instance, methods like R3–R4 are only equivariant in expectation, and works like R4–R6 treat graphs strictly as sequences. In contrast, we highlight that a key challenge in applying SSMs to graphs is that graphs are not inherently sequential. To clarify this distinction, we have revised our paper to include this discussion. Thank you.
>
> **Regarding R5-R10:** We refer the Reviewer to Appendix A, where we discuss GNNs as dynamical systems and recent graph recurrent methods. Following your suggestion, we now cite and highlight the differences with R5–R10 in the revised paper.
>
> **Regarding MixHop:** We refer the Reviewer to the related works section, where we discuss multi-hop architectures like JKNet and DRew. In the revised paper, we added a discussion on MixHop. As shown in Tables 2 and 8, GRAMA outperforms MixHop, and our ablation studies (Tables 9–11) show that even without the selective mechanism, GRAMA still performs better. This highlights that the gain comes not only from selectiveness but also from GRAMA’s overall design: unlike MixHop’s dense, non-recurrent projection, GRAMA uses a recurrent, non-dense aggregation inspired by dynamical systems and modern RNNs. We have added this discussion and further comparisons, including non-selective GRAMA, to the revised paper. The original results are also summarized in the table below.
>
> |Method|Peptides Func|Peptides Struct|Roman-Empire|Amazon-Ratings|OGBN-Arxiv|
> |---|---|---|---|---|---|
> |MixHop-GCN |68.43±0.49|0.2614±0.0023 |79.16±0.70|47.95±0.65|71.29±0.29|
> |GRAMA$_{GCN}$ |70.93±0.78|0.2439±0.0017|88.61±0.43|53.48±0.62|73.86±0.21|
>
>
> **Regarding complexity:**  The Reviewer is correct that GRAMA maintains a feature matrix of size $R \\times n \\times d$, with $R = L$. However, unlike MixHop—which assigns learnable weights per layer and uses a dense layer with space complexity $L \\cdot d^2$—GRAMA is more efficient. In the non-selective (naive) variant, GRAMA learns only $2L$ parameters per block, which is highly scalable since $L \\ll d$ and $L \\ll n$. In the selective variant, parameter count depends only on $d$, with space complexity $d^2$; the $L$ dependence appears only in the $L^2$ complexity of the sequence input, as discussed in Appendix E.3. We have clarified this distinction in the revised paper and, following your suggestion, added a MixHop comparison on OGBN-Arxiv, further demonstrating GRAMA’s effectiveness.
>
>
> **Regarding R13-R15:**  In the original paper, we compare GRAMA with GOAT and Exphormer in Tables 5 and 6, where GRAMA outperforms both. We appreciate the reviewer's suggestion and now include additional comparisons with Polynormer. Together, these results highlight GRAMA’s strong performance relative to other leading methods.
>
> |Method|Roman-empire|Amazon-ratings|Minesweeper|Tolokers|Questions|
> |---|---|---|---|---|---|
> |GOAT|71.59±1.25|44.61±0.50|81.09±1.02|83.11±1.04 |75.76±1.66|
> |Polynormer|92.55±0.3 |54.81±0.49|97.46±0.36 |85.91±0.74| 78.92±0.89|
> |Exphormer|89.03±0.37|53.51±0.46|90.74±0.53|83.77±0.78|73.94±1.06|
> |GRAMA$_{GCN}$ (Ours)|88.61±0.43 |53.48±0.62| 95.27±0.71| 86.23±1.10|79.23±1.16|
> |GRAMA$_{GatedGCN}$ (Ours)|91.82±0.39 |53.71±0.57|98.19±0.58|85.42±0.95|80.47±1.09|
> |GRAMA$_{GPS}$ (Ours) |91.73±0.59|53.36±0.38|98.33±0.55|85.71±0.98|79.11±1.19|
>
> **Regarding runtimes:** As shown in Table 9, GRAMA’s runtime scales proportionally with its backbone. Designed to enhance various GNNs, GRAMA achieves strong performance, especially with backbones like GatedGCN or GPS. We recognize that in some cases, lower computational cost may be important. To address this, one can use the naive ARMA coefficient learning (Section 3.2) or a lighter backbone such as GCN—both of which reduce overhead while still benefiting from GRAMA. Our experiments show these variants maintain strong improvements over their backbones and remain competitive with other methods at lower runtime. We also report GRAMA runtimes with GatedGCN and GPS in the table below and in the revised paper. The key takeaway is that GRAMA offers a flexible and scalable trade-off between efficiency and performance.
>
> |Metho |Depth|4|8|16|32|  |
> |---|---|---|---|---|---|---|
> |GatedGCN|  |27.57|47.98|85.36| 171.27 |  |
> |GPS|  |1139.05| 2286.96  | 4545.46 | OOM |  |
> |GRAMA$_{GatedGCN}$ (Ours)|  |51.49|117.01|270.64|792.32 |  |
> |GRAMA$_{GPS}$ (Ours)|  |1162.13|2346.94|4642.19|OOM|  |

---

> > ### Comment · Reviewer_NJt1 · 2025-04-07
> >
> > I sincerely appreciate the authors’ detailed response, and most of my concerns have been adequately addressed.
> >
> > However, I find that GRAMA performs on par with, or even worse than, attention-based methods on several datasets. This raises an important question: recent graph Transformer variants have demonstrated the ability to model global dependencies within a single layer, often with *linear time and space complexity*. In this context, what is the motivation for adopting a layer-stacking approach (or a step-stacking approach for GRAMA) to expand the receptive field, especially when it introduces *additional computational cost without significant performance gains*—and in some cases, even results in degradation? I suggest the authors include further discussion on this design choice to strengthen the methodological justification of the work.
> >
> > In addition, although GARMA reduces memory consumption by using fewer parameters compared to MixHop, does the parameter reuse across layers lead to out-of-memory (OOM) issues during backpropagation, thereby affecting the scalability of the model? How do the authors address this potential problem?

---

> > > ### Author Response · Authors · 2025-04-07
> > >
> > > We thank the Reviewer for responding to our rebuttal, and for acknowledging that our responses address most of your concerns. Below, we address your questions. We hope that you find them satisfactory and that you will consider revising your score.
> > >
> > > ---
> > >
> > > **Regarding GRAMA vs. attention layers:** Our GRAMA offers a general framework that can be coupled with different GNNs (including MPNNs and Graph Transformers), as shown in our experiments. In particular, this framework allows the achievement of a state space model (SSM) equivalent approach for graph learning tasks that, compared with previous graph SSMs, is (i) permutation-equivariant, (ii) allows working with longer sequences and model interactions that are beyond pairwise interactions at each block.  Therefore, one can also use GRAMA with other backbones, such as the examples suggested by the Reviewer. If the Reviewer has a specific suggestion for such backbones, we are happy to include it in our evaluations.
> > >
> > > **Regarding complexity and performance improvement with GRAMA:** We would like to kindly note again that as shown in our analysis and provided runtimes, while GRAMA adds more computations, which are thoroughly discussed in the paper, it still remains within the magnitude of its backbone. Moreover, we would like to note that **our GRAMA consistently, and often significantly, improves the performance of its backbone, in all of the experiments provided in our paper and rebuttal**. In particular, the design of GRAMA can learn to default to the original backbone. Hence, theoretically (and practically based on our experiments), GRAMA only extends and improves its backbone.  This is also reflected in our discussion with Reviewer c8Wb, and our paper was revised to better reflect that, in addition to other discussions and reasonings on the design of GRAMA.
> > >
> > >
> > > **Regarding "GRAMA performs on par with, or even worse than, attention-based methods on several datasets:"** We kindly note that in our paper and rebuttal, we **consider more than 20 graph attention methods and variants on more than 20 datasets**. In total, we have **132 experimental comparisons with attention-based architectures**. Overall, besides Polynormer which outperforms GRAMA in 3 cases,  **we found our GRAMA to outperform attention-based methods in 129/132 of the comparisons.** Moreover, as discussed above, our method can also be coupled with Polynormer and extend it. We think that, therefore, our GRAMA offers a valuable framework to the graph learning community.
> > >
> > > **For your convenience, we also provide a head-to-head comparison summary of GRAMA with its backbones, showing consistent improvement of the backbones. In any case where GRAMA+Backbone is better than the corresponding Backbone, we mark the result in bold. A broader comparison also exists in our paper in Table 15:**
> > >
> > > | Model | Diameter ($\log_{10}(MSE)\downarrow$) | SSSP ($\log_{10}(MSE)\downarrow$) | Eccentricity ($\log_{10}(MSE)\downarrow$) | Peptides-func (AP $\uparrow$) | Peptides-struct (MAE$\downarrow$) | MalNet-Tiny (Acc $\uparrow$) | Roman-empire (Acc $\uparrow$) | Amazon-ratings (Acc $\uparrow$) | Minesweeper (AUC $\uparrow$) | Tolokers (AUC $\uparrow$) | Questions (AUC $\uparrow$) |
> > > |---|---|---|---|---|---|---|---|---|---|---|---|
> > > | GCN | 0.7424±0.0466 | 0.9499±9.18·10−5 | 0.8468±0.0028 | 59.30±0.23 | 0.3496±0.0013 | 81.00 | 73.69±0.74 | 48.70±0.63 | 89.75±0.52 | 83.64±0.67 | 76.09±1.27 |
> > > | GatedGCN | 0.1348±0.0397 | -3.261±0.0514 | 0.6995±0.0302 | 58.64±0.77 | 0.3420±0.0013 | 92.23±0.65 | 74.46±0.54 | 43.00±0.32 | 87.54±1.22 | 77.31±1.14 | 76.61±1.13 |
> > > | GPS | -0.5121±0.0426 | -3.599±0.1949 | 0.6077±0.0282 | 65.35±0.41 | 0.2500±0.0005 | 92.64±0.78 | 82.00±0.61 | 53.10±0.42 | 90.63±0.67 | 83.71±0.48 | 71.73±1.47 |
> > > | GRAMA$_{GCN}$ (Ours) | **0.2577±0.0368** | **0.0095±0.0877** | **0.6193±0.0441** | **70.93±0.78** | **0.2439±0.0017** | **93.43±0.29** | **88.61±0.43** | **53.48±0.62** | **95.27±0.71** | **86.23±1.10** | **79.23±1.16** |
> > > | GRAMA$_{GatedGCN}$ (Ours) | **-0.5485±0.1489** | **-4.1289±0.0988** | **0.5523±0.0511** | **70.49±0.51** | **0.2459±0.0020** | **93.66±0.40** | **91.82±0.39** | **53.71±0.57** | **98.19±0.58** | **85.42±0.95** | **80.47±1.09** |
> > > | GRAMA$_{GPS}$ (Ours) | **-0.8663±0.0514** | **-3.9349±0.0699** | **-1.3012±0.1258** | **69.83±0.83** | **0.2436±0.0022** | **94.37±0.36** | **91.73±0.59** | **53.36±0.38** | **98.33±0.55** | **85.71±0.98** | **79.11±1.19** |
> > >
> > >
> > > **Regarding memory:** Thank you for the question. In our experiments, we have not encountered OOM issues *if the backbone itself does not yield OOM error*. That is, given an architecture and its configuration that can fit in memory, we are able to use it as a backbone for GRAMA. This is also reflected in our additional runtimes comparison in our responses and in the original paper - **our GRAMA can also be used with deep models without memory issues, given that the backbone model can fit in memory.**

---

### Official Review · Reviewer_tuKb · 2025-03-12

**Overall Recommendation:** 3

**Summary:**

This paper introduces a Graph Adaptive method GRAMA based on a learnable ARMA framework to address limitations in existing Graph State Space Models. GRAMA preserves permutation equivariance while enabling efficient long-range information propagation via a selective attention mechanism. Theoretical connections to Selective SSMs highlight its ability to capture long-range dependencies.

**Claims And Evidence:**

yes

**Essential References Not Discussed:**

no

**Experimental Designs Or Analyses:**

yes

**Methods And Evaluation Criteria:**

yes

**Other Comments Or Suggestions:**

see weakness.

**Other Strengths And Weaknesses:**

### Strengths:
The paper has a strong theoretical foundation with detailed proofs supporting the proposed method. The experimental section is exceptionally thorough, covering a wide range of benchmarks and providing comprehensive comparisons with existing methods.

### Weaknesses:
1. The proposed method relies on three specific GNN backbones: GCN, GatedGCN, and GPS. However, the rationale behind this selection is not clearly discussed. Additionally, the paper does not explore the potential advantages and disadvantages of using other backbones.

2. The comparison with existing methods primarily focuses on performance improvements, but there is limited discussion on the architectural and design differences, especially regarding how the method integrates with graph neural networks.

3. The choice of benchmark tasks may introduce bias, and there is a lack of comparison between graph datasets with varying structural irregularities and heterogeneity, which could affect the generalizability of the results.

4. The paper lacks clear and intuitive visualizations. The method diagram is not straightforward, and there are no direct experimental visuals to illustrate the claimed oversquashing problem, making it harder to connect theory with empirical results.

5. While the appendix is thorough, it is overly lengthy. The core experimental results and conclusions could be distilled more concisely for better clarity and focus.

**Questions For Authors:**

see weakness.

**Relation To Broader Scientific Literature:**

yes

**Theoretical Claims:**

no

---

> ### Author Rebuttal · Authors · 2025-04-01
>
> We thank the Reviewer for acknowledging the **“strong theoretical foundation”** with **“detailed proofs”**, with an **“exceptionally thorough”** experimental section. We would also like to express our gratitude for the thoughtful comments and feedback, to which we have done our best efforts to respond, below. We hope that you find our responses satisfactory and that you will consider revising your score.
>
> ---
>
> **Regarding GNN backbones:**  Thank you for the comment. We selected these backbones because they are widely used and represent both linear MPNNs and graph transformers. Our aim was to demonstrate GRAMA's versatility across backbone types, which is reflected in the consistent improvements over both the GNN backbones and other methods. While linear MPNNs and graph transformers differ in computational complexity, our results show that strong performance is achievable with any of the selected backbones. Overall, our experiments with three popular backbones on 22 datasets—plus 4 more added in the rebuttal in response to Reviewers mg6D and c8Wb—highlight the effectiveness and generality of GRAMA.
>
> **Regarding architectural differences:**  We thank you for the comment. In our paper, we compare both the quantitative results (i.e., downstream performance) of GRAMA and existing methods, primarily focusing on graph SSMs, as well as the qualitative (i.e., architectural and theoretical) differences. We kindly note that in Section 1, Section 2, Section 3, as well as Appendix A,  we state the main differences between GRAMA and other graph SSMs: (i) GRAMA can process sequences which are beyond pairwise interactions, different than the graph SSM  in Huang et al. (2024); and (ii) GRAMA is permutation-equivariant, while other graph SSMs like in Behrouz & Hashemi (2024) and Wang et al. (2024a), are not permutation-equivariant and are based on heuristics that order the graph nodes to obtain a sequence. Nonetheless, following your suggestion, we have revised our paper to better reflect and highlight these differences. We also highlighted differences between other methods like GRIT and GRED, in our response to Reviewer mg6D. Regarding the integration with GNNs, we kindly note that Figure 1 illustrates our method and its ability to integrate with potentially any GNN backbone. This is also reflected by our experiments with different types of backbones such as GCN, GatedGCN, GPS. To fully accommodate your comment, we revised our paper to include and expand the discussions in our response.
>
> **Regarding datasets:**  Thank you for the comment. Our main goal was to evaluate GRAMA’s long-range effectiveness across multiple benchmarks, including Graph Property Prediction (GPP) and LRGB. As noted in Appendix D, GPP graphs are drawn from diverse distributions (e.g., Barabasi-Albert, tree, caveman, line), while LRGB graphs represent molecules with inherently different structures. Additionally, the five heterophilic tasks involve nodes of the same class being distantly connected. These choices ensure benchmark diversity beyond a single application domain. Following suggestions from Reviewers mg6D and c8Wb, and inspired by your comment, we also added experiments on four additional benchmarks (beyond the original 22) to further demonstrate GRAMA’s effectiveness. These results appear in our response to Reviewer c8Wb and have been added to the revised paper.
>
> **Regarding method diagram:**  Thank you for the comment. Figure 1 illustrates GRAMA’s overall approach, showing how a learned selective mechanism aggregates information from previous states alongside a chosen backbone GNN. This process is described in the caption and detailed in Section 3. Following your suggestion, we revised the caption for clarity. Additionally, we refer the Reviewer to Figure 2, which highlights GRAMA’s strong performance on long-range tasks. Inspired by your comment, we now include visualizations of the learned coefficients across datasets to offer further insight into the selective mechanism. Regarding the link between theory and empirical results: (i) Figure 2 shows GRAMA maintains performance as task range increases—consistent with Theorem 4.4; (ii) across long-range benchmarks, GRAMA's competitive results align with the theoretical insights discussed in Section 4. We added this discussion in the revised paper and thank you for helping us improve its clarity.
>
>
> **Regarding Appendix:** Thank you for thoroughly reviewing our appendix and for your comment. Our goal was to provide complete details, including: (i) distinguishing GRAMA from existing methods (Appendix A, related to your comment 2); (ii) theoretical proofs; (iii) implementation details; (iv) experimental settings; (v) additional results and complexity discussion; and (vi) a summary of all results. We appreciate your suggestion and have revised the appendix to present this information more clearly. Thank you.

---

### Official Review · Reviewer_c8Wb · 2025-03-14

**Overall Recommendation:** 5

**Summary:**

This paper introduces a novel GNN architecture based on a state-space model, proposing a new method to transform graph data into a sequence. Unlike previous approaches with similar goals, this work presents a principled approach to sequence generation, ensuring a provably permutation-invariant framework. The proposed model is designed to mitigate oversquashing. Additionally, the paper provides insightful theoretical analysis of the presented architecture.

**Claims And Evidence:**

The claimed contributions of this paper are supported by theoretical results and extensive empirical experiments.

**Essential References Not Discussed:**

The essential references seem to be included, although I am not working in the field of state-space-based GNN models, so I do not have a deep understanding of the existing work.

**Experimental Designs Or Analyses:**

I have checked the experimental design in the main text. There are no apparent soundness or validity issues. The statistical significance of the results should be reported when possible.

**Methods And Evaluation Criteria:**

The chosen synthetic dataset and real dataset are appropriate and sufficient in number to support the claims. They are selected to adequately demonstrate the claimed benefit of the method (ability to model long-range dependencies). Some datasets that are usually reported (e.g., Cluster, CIFAR-10, MNIST) might be missing, but sufficient results are presented.

**Other Comments Or Suggestions:**

- Mentioning guarantees or results related to the Weisfeiler-Lehman test could strengthen this paper. I suspect that it likely retains any guarantees provided by the base GNN layer.

- I would recommend expanding the discussion on complexity, particularly regarding the number of parameters, in the main text rather than relegating it to the supplementary material. While reading, I got the impression that this model would be much more expensive and slower than other GNN architectures, but this is not the case.

**Other Strengths And Weaknesses:**

**Strengths**

- The proposed architecture is novel and principled.
- The work is overall well-written and has great clarity. The experimental section is detailed enough to be reproducible.
- The experimental section is extensive and supports the stated claims.

**Weaknesses**

- Limitations of the proposed architecture are not discussed. One point that could be addressed is the scalability to larger graphs.

**Questions For Authors:**

N/A

**Relation To Broader Scientific Literature:**

The main contribution of this work is to introduce a principled approach to integrating state-space models into a GNN framework. This is important as state-space models are gaining interest due to their attractive properties, and their integration into graph learning is meaningful, given the importance of modeling long-range dependencies in graph settings.

**Theoretical Claims:**

I have not examined the proof of the theoretical results in detail. However, Theorem 4.1 appears to restate known results, and additional citations may be necessary. For example, see [1].
[1] Aoki, M. (1990). State Space and ARMA Models. In M. Aoki (Ed.), State Space Modeling of Time Series (pp. 21–38)

---

> ### Author Rebuttal · Authors · 2025-04-01
>
> We sincerely thank the Reviewer for the positive feedback and assessment of our paper. We are also grateful for the actionable feedback to which we respond below. We have made our best efforts to accommodate each of your comments, and we hope you find our responses satisfactory.
>
> ---
>
> **Regarding benchmarks:** We thank the reviewer for the suggestions for additional datasets. We followed your suggestion (as well as Reviewer mg6D) and now added more results on the mentioned datasets (Cluster, CIFAR10, MNIST, PATTERN). The results, provided in the Table below, show that our GRAMA continues to consistently offer strong performance compared to its backbone GNN as well as other leading methods. We added the results and the discussion to our revised paper.
> | Model | ZINC (exists in paper) | MNIST | CIFAR10 | PATTERN | CLUSTER |
> |---|:---:|:---:|:---:|:---:|:---:|
> | GCN | 0.367±0.011 | 90.705±0.218 | 55.710±0.381  | 71.892±0.334  | 68.498±0.976 |
> | GatedGCN | 0.282±0.015 | 97.340±0.143  | 67.312±0.311  | 85.568±0.088  | 73.840±0.326 |
> | GPS | 0.070±0.004 | 98.051±0.126 | 72.298±0.356  | 86.685±0.059  | 78.016±0.180 |
> | EGT | 0.108±0.009 | 98.173±0.087 | 68.702±0.409  | 86.821±0.020  | 79.232±0.348 |
> | GRIT | 0.059±0.002 | 98.108±0.111  | 76.468±0.881 | 87.196±0.076 | 80.026±0.277 |
> | GRAMA$_{GCN}$ (Ours) | 0.142±0.010 | 97.871±0.188 | 70.283±0.417 | 82.660±0.183 | 74.294±0.595 |
> | GRAMA$_{GatedGCN}$ (Ours) | 0.140±0.008 | 98.119±0.104 | 74.612±0.450 | 86.715±0.099 | 76.883±0.317 |
> | GRAMA$_{GPS}$ (Ours) | 0.061±0.003 | 98.292±0.135 | 75.917±0.408 | 87.406±0.067 | 79.659±0.194 |
>
>
> **Regarding Aoki (1990):** We thank you for the careful reading of our theoretical sections of the paper. In our paper, we cite Aoki (2013) as a reference to background works on SSMs, and in our Theoretical Properties of GRAMA (Section 4), we state that our goal is to adapt findings in the world of SSMs and ARMA models into a graph-learning framework. Our intention is to credit the findings of previous works, as cited and discussed in Section 4 and Appendix B. We thank you for the reference, which we have now added and discussed in our revised paper and appendix. In particular, we have clarified that the results were developed in previous studies for non-graph models, and here we expand it such that it fits into a graph-learning framework.
>
> **Regarding statistical significance:** We thank the Reviewer for acknowledging the soundness and validity of our experiments. Throughout our experiments, we consistently included the standard deviation of our results. In some of the ablation studies, we reported the average performance using the same evaluation settings as in other experiments. In our final version, we will also include the already computed standard deviation of these experiments.
>
> **Regarding limitations:** Thank you for the suggestion. In our paper, we have discussed the complexity as well as runtimes of GRAMA, showing that compared with other methods like graph transformers, it retains better scalability. Following your suggestion below, as well as Reviewer’s mg6D comment, we have revised our paper, to better discuss this aspect in the main paper.
>
> **Regarding WL test:** We welcome your suggestion. Because our model learns to selectively attend to different states, it can also learn to attend to the current state, which means it can default to the backbone GNN. Hence, it can retain at least the expressiveness of the backbone GNN. Due to space limitations, we will include a formal proof in the revised paper. Additionally, we think that an interesting future work will be to understand the exact expressiveness of methods like GRAMA and other graph SSMs in terms of the WL test.
>
> **Regarding complexity discussion:**  We thank the Reviewer for the thoughtful suggestion. We agree that one of the benefits of GRAMA is its reduced computational complexity compared with other models, e.g., transformers. Following your advice, as well as Reveiwer’s mg6D question, we revised our paper such that it includes a broader complexity discussion in the main paper, as well as better referring to Appendix E.3, where we provide a full analysis and runtimes comparison.

---

### Official Review · Reviewer_mg6D · 2025-03-17

**Overall Recommendation:** 3

**Summary:**

This paper introduces GRAMA, which utilizes ARMA (autoregressive moving average) to design graph state space models. The paper claims this way of design can preserve permutation equivariance and enable long-range message passing with good empirical accuracy across multiple datasets.

**Claims And Evidence:**

- The paper needs to better discuss and compare with existing graph mamba works. For example, some claims may not be convincing:
    - "limit their focus to pairwise interactions rather than sequences" (L16) & "however, this design choice may not fully exploit the sequence-handling capacity of SSMs" (L104)
    - Why is it so important to model the input as sequences in graph learning, where things are just not natural sequences?

- There are multiple standard benchmarks for evaluating the capability of capturing long-range dependencies of graph transformers, including datasets such as mnist, cifar10, pattern, cluster, malnet-tiny, PascalVOC-SP,  Peptides-Func, Peptides-Struct, zinc, zinc-full, etc. Even though the paper seems to include many benchmarks, many of them are not that widely used for evaluating graph transformer-like models. It would be better if the authors could include more widely used datasets in this domain.

- The presentation could be improved to better show why ARAMA is a better graph SSM compared with existing methods. What makes ARMA so special? And more baselines should be included to justify the superiority of ARAMA, e.g., [1, 2, 3]

[1] Ma, Liheng, et al. "Graph inductive biases in transformers without message passing." International Conference on Machine Learning. PMLR, 2023.

[2] Ding, Yuhui, et al. "Recurrent distance filtering for graph representation learning." arXiv preprint arXiv:2312.01538 (2023).

[3] Huang, Yinan, Siqi Miao, and Pan Li. "What Can We Learn from State Space Models for Machine Learning on Graphs?." arXiv preprint arXiv:2406.05815 (2024).

**Essential References Not Discussed:**

See above.

**Experimental Designs Or Analyses:**

See above.

**Methods And Evaluation Criteria:**

See above.

**Other Comments Or Suggestions:**

See above.

**Other Strengths And Weaknesses:**

Even though some included datasets are less popular, it's good to see GRAMA can perform well on those datasets.

**Questions For Authors:**

Another benefit that SSMs can bring compared with transformers is efficiency. There is some preliminary timing in the appendix, but how ARAMA scale to larger graphs? Can it inherit the efficiency from SSMs?

**Relation To Broader Scientific Literature:**

This paper is related to graph state space models and graph transformers, which may better capture long-range dependencies in graphs. This paper is especially related to graph mamba, which finds ways to translate graphs into sequences and then apply SSMs.

**Theoretical Claims:**

I did not go over the appendix, but the theoretical claims in the main text look reasonable as many of them are from the SSMs.

---

> ### Author Rebuttal · Authors · 2025-04-01
>
> We thank the Reviewer for the thoughtful comments and the actionable feedback. We have taken significant measures to accommodate each of your comments. We hope that you will find our responses satisfactory, and that you will consider raising your score. We are also happy to read that the Reviewer acknowledges that **“GRAMA can perform well”**.
>
> ---
>
> **Regarding sequences in graph learning:** Thank you for the question. Indeed, graphs are by design not sequences, as discussed in the Introduction section. However, in order to utilize mechanisms such as SSMs or ARMA methods, to capture long-range dependencies, one needs to process sequences that are longer than the length of 2 (i.e., pairwise interactions). This is also reflected in existing literature such as Behrouz & Hashemi (2024) and Wang et al. (2024a). However, as we discuss throughout the paper, their shortcoming is that they rely on heuristics of ordering the nodes, which is not data-driven, and also not permutation equivariant. Following your question, we refined the discussion to better reflect this.
>
> **Regarding benchmarks:**  We thank the reviewer for recognizing the large number of datasets and benchmarks considered in our paper, which covers 22 synthetic and real-world datasets. Moreover, following your suggestion, we now added results on the datasets suggested by the reviewer, from [1]. Below, we provide a table with these results, as well as a comparison with other methods. As can be seen, our GRAMA offers competitive performance, compared with other methods, while maintaining the complexity of the backbone GNN.
> | Model | ZINC (our results exist in paper) | MNIST | CIFAR10 | PATTERN | CLUSTER |
> |---|:---:|:---:|:---:|:---:|:---:|
> | GCN | 0.367±0.011 | 90.705±0.218 | 55.710±0.381  | 71.892±0.334  | 68.498±0.976 |
> | GatedGCN | 0.282±0.015 | 97.340±0.143  | 67.312±0.311  | 85.568±0.088  | 73.840±0.326 |
> | GPS | 0.070±0.004 | 98.051±0.126 | 72.298±0.356  | 86.685±0.059  | 78.016±0.180 |
> | EGT | 0.108±0.009 | 98.173±0.087 | 68.702±0.409  | 86.821±0.020  | 79.232±0.348 |
> | GRIT | 0.059±0.002 | 98.108±0.111  | 76.468±0.881 | 87.196±0.076 | 80.026±0.277 |
> | GRAMA$_{GCN}$ (Ours) | 0.142±0.010 | 97.871±0.188 | 70.283±0.417 | 82.660±0.183 | 74.294±0.595 |
> | GRAMA$_{GatedGCN}$ (Ours) | 0.140±0.008 | 98.119±0.104 | 74.612±0.450 | 86.715±0.099 | 76.883±0.317 |
> | GRAMA$_{GPS}$ (Ours) | 0.061±0.003 | 98.292±0.135 | 75.917±0.408 | 87.406±0.067 | 79.659±0.194 |
>
> **Regarding GRAMA and other methods:** Thank you for the question. In our paper, we attribute GRAMA’s improvements over existing graph SSMs and related methods to three key differences: (i) GRAMA supports beyond-pairwise interactions while preserving permutation equivariance—unlike [3], Behrouz & Hashemi (2024), and Wang et al. (2024); (ii) [2] employs LRU without a selective mechanism, which our results (Tables 9–11) show to be impactful; and (iii) GRIT [1], a graph transformer, differs fundamentally from graph SSMs in both computational cost and operation. GRIT emphasizes expressive positional encodings and is more resource-intensive than GraphGPS, whereas GRAMA is efficient, permutation-equivariant, and designed for long-range propagation.
>
> **Regarding more baselines:** We kindly note that in our comparisons, we have considered 22 datasets, as well as more than 30 baselines (i.e., different methods). In all cases, we see that our GRAMA offers similar or better performance compared with other methods, while maintaining the complexity of the backbone GNN. Nonetheless, we welcome your suggestion, and we now compare, discuss, and cite the works mentioned by the Reviewer in our revised paper. We also provide the results in the Table below, showing the competitive performance offered by our GRAMA:
> | Model           | Peptide Func ($\uparrow$) | Peptide Struct ($\downarrow$) |
> |----------------|:---------------------:|:----------------------:|
> | GRIT [1]          | 0.6988 ± 0.0082       | 0.2460 ± 0.0012        |
> | GRED  [2]         | 0.7085 ± 0.0027       | 0.2503 ± 0.0019        |
> | GRED + LapPE [2]   | 0.7133 ± 0.0011 | 0.2455 ± 0.0013        |
> | GSSC [3]          | 0.7081 ± 0.0062       | 0.2459 ± 0.0020        |
> | GRAMA (Ours, best variant)          | 0.7093 ± 0.0078       | 0.2436 ± 0.0022  |
>
>
> **Regarding efficiency:** We thank you for the important question. The Reviewer is correct that one of the advantages of SSMs is their efficiency compared with transformers. In our paper, we provided the training and inference runtimes of GRAMA, including a comparison with other types of methods, from linear MPNNs (GCN and GatedGCN), to graph transformers (GPS). These results are shown in Table 9 and Table 10 in Appendix E.3. Furthermore, we have also discussed the theoretical complexity of GRAMA in Appendix E.3. Following your question, as well as Reviewer’s c8Wb suggestion, we moved the main discussion to the main paper, and we better refer to Appendix E.3 from the main paper.

---

> > ### Comment · Reviewer_mg6D · 2025-04-09
> >
> > I thank the authors for their detailed responses, which addressed most of my concerns and these results should be included in the revised manuscript. I find that the proposed method can still be interesting for people studying sequence models on graphs. Therefore, I will increase my rating.

---

> > > ### Author Response · Authors · 2025-04-09
> > >
> > > We thank the Reviewer for the response to our rebuttal and for increasing their rating. We added the results and discussions provided in our responses to the revised paper, and we think that the constructive feedback helped us to improve the paper. Thank you.
> > >
> > >
> > > With warm regards,
> > >
> > > Authors.

---

### Decision · Program_Chairs · 2025-05-01

**Decision:**

Accept (spotlight poster)

**Comment:**

This paper proposes a graph state space model based on an autoregressive moving average framework. Overall, all reviewers acknowledged that the technical contributions and empirical results are solid. The main concerns raised were: (1) limited discussion of related state space models (SSMs) or sequence models for graphs, (2) missing results on some widely-used benchmarks, (3) lack of comparisons with key baselines, and (4) insufficient justification for certain architectural choices.

The authors’ rebuttal effectively addressed most of these concerns, leading to a consensus among the reviewers in favor of acceptance. After carefully reading the reviews and discussion, the AC agrees with this assessment and recommends acceptance.